# ADVANCING COMPLEX VIDEO OBJECT SEGMENTATION VIA PROGRESSIVE CONCEPT CONSTRUCTION

**Zhixiong Zhang**[1,2,3*] **Shuangrui Ding**[4*] **Xiaoyi Dong**[3,4†] **Songxin He**[5] **Jianfan Lin**[5]
**Junsong Tang**[5] **Yuhang Zang**[3] **Yuhang Cao**[3] **Dahua Lin**[4,6,3] **Jiaqi Wang**[2,3†]

[1]Shanghai Jiao Tong University    [2]Shanghai Innovation Institute
[3]Shanghai Artificial Intelligence Laboratory    [4]The Chinese University of Hong Kong
[5]Harbin Institute of Technology    [6]CPII under InnoHK

`https://rookiexiong7.github.io/projects/SeC/`

## ABSTRACT

We propose Segment Concept (SeC), a concept-driven video object segmentation (VOS) framework that shifts from conventional feature matching to the progressive construction and utilization of high-level, object-centric representations. SeC employs Large Vision-Language Models (LVLMs) to integrate visual cues across diverse frames, constructing robust conceptual priors. To balance semantic reasoning with computational overhead, SeC forwards the LVLMs only when a new scene appears, injecting concept-level features at those points. To rigorously assess VOS methods in scenarios demanding high-level conceptual reasoning and robust semantic understanding, we introduce the Semantic Complex Scenarios Video Object Segmentation benchmark (SeCVOS). SeCVOS comprises 160 manually annotated multi-scenario videos designed to challenge models with substantial appearance variations and dynamic scene transformations. Empirical evaluations demonstrate that SeC substantially outperforms state-of-the-art approaches, including SAM 2 and its advanced variants, on both SeCVOS and standard VOS benchmarks. In particular, SeC achieves an 11.8-point improvement over SAM 2.1 on SeCVOS, establishing a new state-of-the-art in concept-aware VOS. The code, checkpoint and benchmark are open-sourced here.

## 1 INTRODUCTION

Video Object Segmentation (VOS) is a pivotal task in computer vision, focusing on the precise delineation and temporal tracking of target objects within video sequences. By capturing both spatial and temporal dynamics, VOS enables comprehensive scene understanding, which is essential for a range of applications including autonomous driving (Siam et al., 2021), robotic perception (Griffin et al., 2020), video editing (Tu et al., 2025), and intelligent surveillance systems (Ammar et al., 2019). A core component of mainstream VOS models (Ravi et al., 2025; Cheng & Schwing, 2022; Zhou et al., 2024) is memory-based matching, where the target in each frame is identified by measuring its pixel-level similarity to previously observed instances. This approach achieves solid performance on standard VOS benchmarks (Pont-Tuset et al., 2017; Xu et al., 2018).

Despite their success, we argue that these methods remain far inferior to human capability in real-world scenarios, particularly when the appearance of the target changes drastically across frames due to occlusions, viewpoint shifts, or complex scenes. We think that this limitation arises from a fundamental gap between how machines and humans perceive objects over time. Human perception is not confined to surface-level similarity; instead, it involves the construction of a holistic, conceptual understanding of the target object by integrating observations across frames. This high-level representation, which we refer to as an **object-level concept**, allows humans to robustly recognize the same object even under significant appearance or scene variations. Take Figure 1(a) as an example: although the target (Harry Potter) remains visually consistent with his red and gold uniform,

---

*Equal Contribution
†Corresponding Author

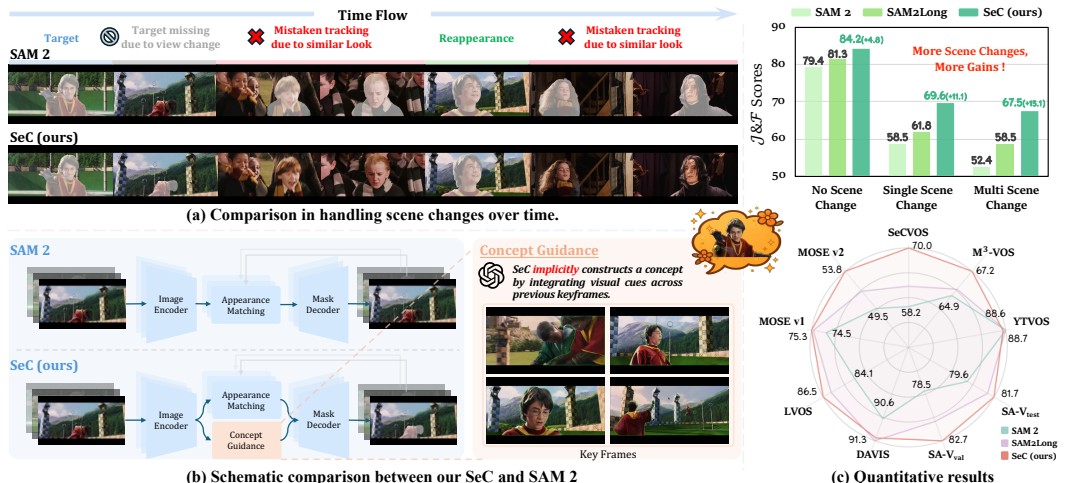

Figure 1: Overview of our Segment Concept (SeC) framework. **(a)** Compared to SAM 2, our model maintains better target tracking under severe appearance changes and scene transitions by leveraging concept-level guidance. **(b)** Schematic comparison between SeC and SAM 2. SeC integrates both low-level appearance matching and high-level concept priors. **(c)** Quantitative results show that SeC consistently outperforms strong baselines, especially in scenarios involving multiple scene changes.

previous VOS model like SAM 2 (Ravi et al., 2025) frequently loses track of him when the scene changes or other characters with similar appearances are introduced. However, if the model were capable of concept-level reasoning, for example by recognizing that Harry is an active player rather than a spectator, such errors could be significantly reduced.

This observation motivates a paradigm shift: from conventional appearance matching to concept-driven segmentation. We take a step in this direction by equipping segmentation models with the ability to form and leverage high-level object concepts over time. To achieve this, we introduce **Se**gment **C**oncept (**SeC**), a concept-driven segmentation framework that progressively constructs a concept-level representation of the target object by integrating information across frames. Rather than relying on superficial appearance matching, SeC leverages the conceptual reasoning capabilities of large vision-language models (LVLMs), drawing upon their rich visual understanding and vast knowledge to build and refine object-level concepts. This enables robust segmentation under challenging conditions such as occlusions, appearance changes, and scene variations. Specifically, SeC samples a representative subset of past frames to serve as input to the LVLM. These keyframes, arranged in temporal order along with the current query frame, are processed by the LVLM, which uses a learnable concept token to distill the concept essence of the target. Note that we only extract the hidden embedding of this token, making the LVLM usage lightweight without generating any additional text. This semantic representation is then injected into the query-frame feature via cross-attention, guiding segmentation with conceptual priors rather than relying solely on low-level features. We show that SeC can progressively model the concept of the target object on the fly, and its performance further improves when the construction is switched to offline mode. This highlights that leveraging LVLM-derived object-level features is beneficial for object referring.

To avoid frequent calls to LVLMs and unfriendly computation cost in the online mode, we draw inspiration from human behavior: for most coherent frames, quick glances are sufficient; only when significant changes occur, such as occlusions or abrupt shifts, do we rely on deeper reasoning with previously formed concepts to re-identify the target. To this end, SeC further employs a scene-adaptive activation strategy: it invokes LVLM-based concept reasoning when complex variations arise, updating the concept representation accordingly. For simpler, stable scenes, it falls back to an enhanced matching mechanism for efficient segmentation. This switch-mode design yields an online segmentation pipeline that is both robust to complex dynamics and computationally efficient.

To better benchmark our model's concept-level reasoning capabilities against prior work, we carefully curate the **Se**mantic **C**omplex Scenarios **V**ideo **O**bject **S**egmentation benchmark (**SeCVOS**). SeCVOS consists of 160 manually annotated multi-shot videos, selected from the Shot2Story

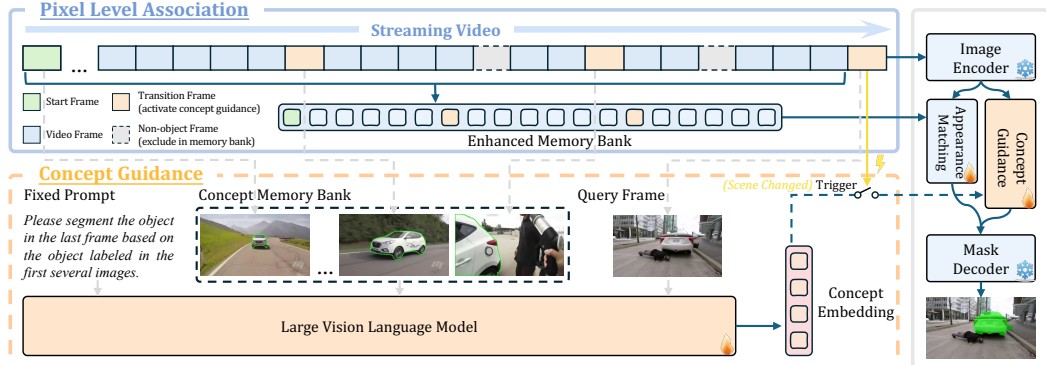

Figure 2: The architecture of our proposed SeC framework. For temporally coherent frames, it relies on Pixel Level Association **(Top)** for efficient, memory-based tracking. Upon a scene change, it activates the Concept Guidance module **(Bottom)**, which leverages a LVLM to build a high-level concept of the target object that is then fused with visual features to guide the segmentation.

dataset (Han et al., 2025) and additional videos crawled from YouTube. To the best of our knowledge, SeCVOS exhibits the highest average number of scenes and the highest disappearance rate among existing VOS benchmarks. Highly discontinuous frame sequences with frequent object re-appearances and dynamic visual changes pose significant challenges to existing VOS methods. Experimental evaluations demonstrate that state-of-the-art memory-based models such as Cutie (Cheng et al., 2024) and SAM 2 (Ravi et al., 2025) achieve limited success on SeCVOS, all scoring below 65 $\mathcal{J}\&\mathcal{F}$, highlighting the necessity for improved semantic reasoning capabilities in current VOS approaches. We plan to open-source SeCVOS benchmark to facilitate further advancements in semantic-level video object segmentation.

Moreover, extensive evaluations across 8 VOS benchmarks validate the effectiveness of our SeC framework. On the challenging SeCVOS benchmark, our method significantly outperforms SAM 2.1 and its recent variants, achieving an average improvement of 11.8 points in $\mathcal{J}\&\mathcal{F}$ over SAM 2.1. Besides, SeC consistently surpasses prior state-of-the-art across 5 standard benchmarks. Specifically, it improves over SAM 2.1 by 4.1 on SA-V (Ravi et al., 2025), 4.3 on MOSE v2 (Ding et al., 2025a), and 2.4 on LVOS v2 (Hong et al., 2024). This demonstrates the advantage of integrating fine-grained pixel association with object-level semantic reasoning derived from multimodal LLMs.

## 2 RELATED WORK

**Memory-based VOS.** VOS models typically propagate labels by matching pixel-level features between query and memory frames. Classical memory-based models (Oh et al., 2019; Cheng et al., 2023; Zhou et al., 2024; Duke et al., 2021; Liang et al., 2020; Oh et al., 2018; Seong et al., 2020; Cheng & Schwing, 2022; Ding et al., 2024; Xie et al., 2021; Yang & Yang, 2022; Yang et al., 2021; Qian et al., 2023; Ding et al., 2022) perform well on short-term tracking but often struggle with distractors due to their reliance on low-level visual cues. Recent methods incorporate object-level information to improve robustness (Athar et al., 2022; Wang et al., 2023; Cheng et al., 2024; Liu et al., 2025). For instance, Cutie (Cheng et al., 2024) introduces object-level memory queries that encode semantic and long-term context, enabling stronger target-background separation. ISVOS (Wang et al., 2023) injects features from a pre-trained Mask2Former (Cheng et al., 2022) detector to make embeddings instance-aware. Furthermore, recent unified segmentation frameworks (Athar et al., 2023; Yan et al., 2023; Li et al., 2024) have achieved strong performance on VOS by jointly modeling multiple tasks. While both models show the benefits of adding semantic cues, their semantic reasoning remains limited to instance-level features. In our work, we leverage LVLMs to inject rich concept-level semantic features into the memory module, further strengthening the model's semantic understanding.

**LVLMs for fine-grained perception.** Large vision-language models (LVLMs) (Hurst et al., 2024; Team et al., 2024; Xing et al., 2025b;a; Chen et al., 2024c;a;d; Dong et al., 2026; Qian et al., 2025;

Ding et al., 2025b;c; Zhao et al., 2025; Wei et al., 2026; 2025; Zhang et al., 2025a) have recently emerged as powerful tools for bringing semantic understanding into dense prediction tasks (Lin et al., 2025; Lai et al., 2024; Yan et al., 2024; Bai et al., 2024; Yuan et al., 2025; Tang et al., 2025; Li et al., 2026). LISA (Lai et al., 2024) pioneered reasoning-based segmentation for images by using an LVLMs with a special [SEG] token that is decoded into a mask. VISA (Yan et al., 2024) extends this concept to videos by integrating text-guided keyframe selection with a SAM-style decoder for per-frame segmentation. UFO (Tang et al., 2025) takes this methodology a step further by unifying detection, segmentation, and captioning tasks through an open-ended language interface. In contrast to these text-driven paradigms, our work focuses on implicitly leveraging the conceptual reasoning capacity of LVLMs, without any explicit textual reasoning. We repurpose the LVLM as a visual concept extractor to guide segmentation directly through latent object-level reasoning.

**VOS benchmarks.** Several recent datasets (Li et al., 2013; Ochs et al., 2013; Xu et al., 2018; Ding et al., 2023; Hong et al., 2024; Ravi et al., 2025; Chen et al., 2024b) have pushed VOS evaluation toward more challenging settings. MOSE (Ding et al., 2023) introduces complex real-world scenes with frequent occlusions, crowded backgrounds, and disappearing-reappearing targets, exposing failure cases where traditional models struggle. SA-V (Ravi et al., 2025) scales up to a massive dataset of ∼51k videos, including small, occluded, and reappearing objects to evaluate mask propagation. Meanwhile, LVOS (Hong et al., 2024) focuses on long-term segmentation: its videos average over 60 seconds and feature long-duration object interactions such as objects leaving and later re-entering the scene. Notably, none incorporate multi-view scenarios or concept-level variation, making it difficult to assess a model's higher-level semantic perception or reasoning capabilities. In contrast, our proposed benchmark SeCVOS is designed to fill this gap. It includes complex multi-shot contextual changes throughout the sequence. This setup requires models to go beyond low-level tracking. They must reason about the target's identity, roles, and intent as the contextual shifts, effectively evaluating semantic understanding in video object segmentation.

## 3  METHOD

### 3.1  PRELIMINARY STUDY ON CURRENT VOS

To understand the limitations of current VOS approaches in complex scenarios, we conduct a detailed evaluation on our SeCVOS benchmark. As shown in Figure 1(c), we categorize videos by the number of scene transitions and report the standard metric $\mathcal{J}\&\mathcal{F}$. Surprisingly, even the state-of-the-art SAM 2 model (Ravi et al., 2025) exhibits substantial performance degradation in videos with only one scene changes. These results indicate the limitations of memory-based designs that rely heavily on low-level visual similarity, lacking the conceptual reasoning needed to maintain object identity across drastic appearance variations.

In contrast, recent LVLMs (Hurst et al., 2024; Guo et al., 2025; Chen et al., 2024d; Wang et al., 2024) have demonstrated impressive visual understanding and reasoning capabilities. Given a sequence of reference frames and a query frame with significant appearance or scene changes, LVLMs can correctly localize the target with reasonable justifications.

This suggests that LVLMs possess the ability to infer object identity beyond surface-level cues, by leveraging powerful visual perception and conceptual reasoning grounded in vast multimodal knowledge. Inspired by this, we propose SeC, a novel framework that integrates LVLM-based object concepts into the video segmentation pipeline. Our model demonstrates strong robustness against drastic scene variations, a major limitation of prior VOS methods.

### 3.2  SEGEMENT CONCEPT MODEL

The architecture of our proposed framework is depicted in Figure 2. Our goal is to enhance a VOS model with concept-level LVLM-based guidance, which enables the learning of object-level representations that are robust to significant appearance changes. At the same time, the model retains the ability to provide reliable pixel-level guidance when no visual scene change is detected.

**Concept guidance with an LVLM.** To facilitate robust concept-level reasoning, we maintain a sparse keyframe bank throughout the video, which provides a diverse view of the target concept to a large vision-language model (LVLM). This bank is initialized with the first annotated frame and

Table 1: Ablation on concept guidance. The offline mode constructs a more holistic concept of the target object.

| Concept construction | $\mathcal{J}\&\mathcal{F}$ | $\mathcal{J}$ | $\mathcal{F}$ |
|---|---|---|---|
| None | 62.2 | 61.8 | 62.6 |
| Online | 70.0 | 69.7 | 70.2 |
| Offline | 71.8 | 71.5 | 72.1 |

Table 2: Efficiency comparison of SeC and SAM 2. SeC achieves superior performance at a comparable throughput, benchmarked on one NVIDIA A800 GPU.

| Benchmark | Method | $\mathcal{J}\&\mathcal{F}$ | Con. Guid. Ratio (%) | Throughput $(s^{-1})$ |
|---|---|---|---|---|
| SeCVOS | SeC | 70.0 | 7.4 | 14.8 |
| | SAM 2 | 58.2 | N/A | 22.0 |
| SA-V | SeC | 82.7 | 1.0 | 18.1 |
| | SAM 2 | 78.6 | N/A | 22.0 |

dynamically updated during tracking. A new frame is added when it both differs significantly from existing keyframes and yields a confident segmentation result, ensuring diversity without sacrificing reliability. To balance efficiency and semantic coverage, we retain only the initial frame and a FIFO buffer of the most recent representative keyframes, capped by a fixed window size. This ensures that the LVLM receives a compact yet semantically rich set of frames for robust concept distillation. Inspired by LISA (Lai et al., 2024), we append a special <SEG> token to the end of the keyframe sequence, prompting the LVLM to summarize the object concept into this special token. The hidden state corresponding to the <SEG> token is then extracted as the object-level concept guidance vector. This method allows the model to implicitly build the concept within a single, efficient forward pass, rather than performing auto-regressive decoding to generate explicit textual reasoning.

**Scene-adaptive activation strategy.** Since most consecutive frames exhibit high temporal coherence, applying concept-level guidance to every frame is computationally redundant. Instead, lightweight pixel-level matching suffices in these cases. To this end, we propose a scene-adaptive activation strategy. Specifically, we detect whether the incoming frame exhibits a significant scene change compared to the previous one. If no such change is detected, we rely solely on the pixel-level association memory and feed the memory-enhanced image features directly into the mask decoder to generate the final prediction. Otherwise, we activate concept-level reasoning via the LVLM. The resulting concept vector is fused with the current frame features through a lightweight cross-attention module. The concept-enhanced spatial features are then pointwise added to the memory-enhanced features, enabling the model to produce segmentation predictions guided by both semantic priors and low-level visual correspondence. This fusion effectively combines high-level semantic concept priors from the LVLM with fine-grained pixel visual cues, enabling the model to remain robust and efficient across drastic appearance and scene variations.

### 3.3 DISCUSSION

In this section, we present a two-part practical analysis to shed light on the intuition behind SeC.

**Does SeC progressively construct concept-level representation?** During the online video segmentation process, frames are segmented sequentially, and the object concept is incrementally constructed as the video progresses. As a result, the final concept obtained after processing the entire video can be considered an expressive representation. This naturally leads to an intuitive idea: if the concept is indeed refined progressively, re-segmenting the video using the finalized concept should yield improved results. To validate this hypothesis, we define this re-segmentation process as an "offline" segmentation task and evaluate its effectiveness on the SeCVOS benchmark.

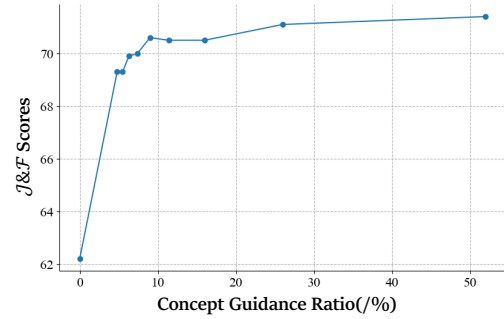

Figure 3: $\mathcal{J}\&\mathcal{F}$ Curve in terms of concept guidance ratio on SeCVOS. Sparse activation (*e.g.*, under 10%) achieves strong performance.

As shown in Table 1, the offline strategy yields the highest performance, indicating that concept representations constructed from a more diverse and comprehensive set of frames lead to better segmentation quality. This aligns well with our core intuition: the model benefits from observing a richer set of visual cues to form a more complete and robust understanding of the target object.

**Does SeC require frequent concept guidance?** To investigate the optimal frequency for activating LVLM-based concept reasoning in SeC, we conduct an ablation study on SeCVOS, varying the concept activation rate. This is implemented by adjusting the threshold used to determine whether a scene change has occurred. As illustrated in Figure 3, enabling concept guidance on fewer than 10% of frames already results in a significant improvement in segmentation performance, with marginal gains beyond that point. This observation suggests that frequent activation of concept-level reasoning is unnecessary. Sparse yet timely activations are sufficient to capture critical semantic transitions.

Furthermore, Table 2 highlights that SeC maintains a competitive inference speed despite the additional reasoning cost. On both SeCVOS and SA-V benchmarks, SeC achieves higher $\mathcal{J}\&\mathcal{F}$ scores with minimal concept guidance usage (7.4% and 1.0%, respectively). This confirms that our scene-adaptive activation strategy effectively balances accuracy and efficiency, selectively invoking concept reasoning only when appearance variations demand it.

# 4    SeCVOS Benchmark

Benchmarks play a crucial role in driving model breakthroughs by providing standardized evaluation protocols. However, we observe that most existing VOS benchmarks (Ding et al., 2023; Ravi et al., 2025; Hong et al., 2023; Ochs et al., 2013; Li et al., 2013) are becoming saturated, with state-of-the-art models already achieving over 90 in $\mathcal{J}\&\mathcal{F}$ scores on widely-used datasets such as YouTube-VOS (Xu et al., 2018) and DAVIS (Pont-

Table 3: Comparison between our SeCVOS and existing VOS benchmarks in terms of videos count, average duration, disappearance rate and number of scenes[1].

| VOS Benchmark | #Videos | Duration(s) Avg. | Disapp. Rate | #Scenes Avg. |
|---|---|---|---|---|
| DAVIS | 90 | 2.87 | 16.1% | 1.06 |
| YTVOS | 507 | 4.51 | 13.0% | 1.03 |
| MOSE | 311 | 8.68[2] | 28.8% | 1.06 |
| SA-V | 155 | 17.24 | 25.5% | 1.09 |
| LVOS | 140 | 78.36 | 7.8% | 1.47 |
| **SeCVOS (ours)** | 160 | 29.36 | **30.2%** | **4.26** |

Tuset et al., 2017). As a result, further improvements on these benchmarks offer diminishing insights into model robustness. More critically, current benchmarks fail to incorporate dedicated evaluation settings that assess a model's performance under semantically challenging conditions, such as long-range occlusions, scene discontinuities, and cross-shot object reasoning.

To address this gap, we propose the **Se**mantic **C**omplex Scenarios **V**ideo **O**bject **S**egmentation (**SeCVOS**) benchmark, specifically designed to assess a model's ability to perform high-level semantic reasoning across complex visual narratives. SeCVOS contains 160 carefully curated multi-shot videos characterized by: 1) Highly discontinuous frame sequences, 2) Frequent reappearance of objects across disparate scenes, and 3) Abrupt shot transitions and dynamic camera motion.

These characteristics introduce substantial challenges for existing memory-based approaches, which predominantly rely on local visual similarity and often fail to maintain object identity across shots. Despite the challenges within these scenarios, they are frequently encountered in real-world VOS applications, such as video editing, surveillance, and story-centric content understanding. Therefore, developing benchmarks that target these conditions is both necessary and important.

To construct the SeCVOS benchmark, we first filtered videos with three criteria to ensure sufficient spatiotemporal complexity: (1) a minimum duration of 20 seconds, (2) semantically meaningful. The semantics are filtered following the strategy introduced in the Shot2Story (Han et al., 2025) to remove less informative videos. Next, we employed GPT-4o to analyze the video content and identify target objects that appear frequently and unambiguously across scenes. Initial object masks were generated using SAM 2 (Ravi et al., 2025), and subsequently refined through multiple rounds of manual correction to ensure high-quality and accurate annotations.

The resulting SeCVOS benchmark consists of 160 multi-shot videos, each averaging 29.36 seconds in duration and containing 4.26 distinct scenes per video, significantly surpassing existing benchmarks in scene diversity. As shown in Table 3, SeCVOS features a high disappearance rate of 30.2%, reflecting the frequent occlusions and reappearances of objects across shots. In contrast, prior benchmarks contain mostly single-scene with low semantic discontinuity. Further details about the SeCVOS benchmark are provided in the Appendix B.

---

[1]Scene counts are consistently estimated using the `scenedetect` library.
[2]Estimated using 6 FPS for MOSE.

Table 4: Performance comparison with prior work on the SeCVOS benchmark, demonstrating better robustness of our SeC to drastic appearance and scene variations.

| Method | No Scene Change | | | Single Scene Change | | | Multi Scene Change | | | Overall |
|---|---|---|---|---|---|---|---|---|---|---|
| | $\mathcal{J}\&\mathcal{F}$ | $\mathcal{J}$ | $\mathcal{F}$ | $\mathcal{J}\&\mathcal{F}$ | $\mathcal{J}$ | $\mathcal{F}$ | $\mathcal{J}\&\mathcal{F}$ | $\mathcal{J}$ | $\mathcal{F}$ | $\mathcal{J}\&\mathcal{F}$ |
| Xmem (Cheng & Schwing, 2022) | 71.9 | 72.0 | 71.8 | 47.0 | 47.9 | 46.2 | 41.9 | 42.4 | 41.4 | 48.4 |
| DEVA (Cheng et al., 2023) | 71.6 | 71.6 | 71.5 | 48.5 | 48.4 | 48.6 | 46.4 | 46.0 | 46.8 | 49.7 |
| Cutie-base (Cheng et al., 2024) | 72.5 | 72.2 | 72.8 | 53.0 | 52.9 | 53.2 | 48.3 | 47.8 | 48.9 | 52.7 |
| SAM2.1 (Ravi et al., 2025) | 79.4 | 79.1 | 79.7 | 58.5 | 58.2 | 58.8 | 52.4 | 52.1 | 52.6 | 58.2 |
| SAMURAI (Yang et al., 2024) | 81.8 | 81.6 | 81.9 | 60.6 | 60.6 | 60.7 | 59.3 | 58.9 | 59.7 | 62.2 |
| SAM2.1Long (Ding et al., 2025d) | 81.3 | 81.0 | 81.6 | 61.8 | 61.6 | 62.0 | 58.5 | 58.1 | 58.9 | 62.3 |
| **SeC (Ours)** | **84.2** +4.8 | **83.8** | **84.5** | **69.6** +11.1 | **69.5** | **69.7** | **67.5** +15.1 | **67.0** | **68.0** | **70.0** +11.8 |

# 5 EXPERIMENTS

## 5.1 IMPLEMENTATION DETAILS

**Model Architecture.** Our model is built upon the SAM 2.1-large backbone (Ravi et al., 2025), reusing its its image encoder and mask decoder components without fine-tuning. On top of this base, we incorporate a pixel-level association memory and an LVLM-based concept guidance module to enhance temporal modeling and semantic reasoning.

In practice, we adopt the memory attention mechanism of SAM 2 as the foundation for our pixel-level association memory. On top of this, we augment the memory module with an enhanced long-term memory by extending the temporal positional encoding to support a wider temporal window of up to 22 frames. Following SAM2Long (Ding et al., 2025d), we apply an object-aware filtering strategy that picks only frames with non-zero occlusion scores, ensuring that memory is constructed from frames where a visible object is present. This ensures that memory is both temporally broad and semantically relevant, reducing noise from uninformative frames. For the concept guidance module, we employ InternVL 2.5 (Chen et al., 2024d) as the base model. The final model is achieved through a two-stage training process that first establishes long-term temporal modeling before fine-tuning the LVLM for concept-level reasoning. Further training details are provided in the Appendix A.

**Scene change detection.** To determine whether a frame should trigger concept-level reasoning, we employ a lightweight HSV-based scene change detector. Specifically, we compute 2D color histograms over the hue and saturation channels of the current and previous frames, and measure their difference using the Bhattacharyya distance. A scene change is detected if the distance exceeds a predefined threshold, which we set to 0.35 by default. Empirically, we find this threshold to be robust against minor variations while remaining sensitive to significant appearance shifts.

**Benchmarks.** To evaluate our method, we conduct experiments on seven standard video object segmentation (VOS) benchmarks: SA-V (Ravi et al., 2025), LVOS v2 (Hong et al., 2024), MOSE v1 (Ding et al., 2023), DAVIS (Pont-Tuset et al., 2017), YouTube-VOS (Xu et al., 2018), $M^3$-VOS (Chen et al., 2025), MOSE v2 (Ding et al., 2025a) and our proposed SeCVOS dataset. We follow the standard evaluation protocol for each benchmark and report its primary metrics. All evaluations are performed under the semi-supervised setting, where the ground-truth mask of the first frame is provided. Further details of these benchmarks can be found in the Appendix D.

## 5.2 MAIN RESULTS ON SeCVOS

We present the performance comparison on the SeCVOS benchmark in Table 4. Our approach consistently outperforms prior art across various settings, including no scene transition, single-scene, and multi-scene scenarios. Notably, as the number of scene transitions increases, the performance gap between our method and prior approaches becomes larger. Even Cutie (Cheng et al., 2024), which claims to leverage object-level representations for improved tracking, fails to maintain performance on SeCVOS. This aligns with our hypothesis that previous VOS methods largely rely on superficial object appearance cues and lack the capacity to form robust, concept-level understanding. This verifies our integration of LVLM-based concept reasoning into the segmentation pipeline enables the model to effectively distill object-level concepts across diverse and discontinuous scenes.

Table 5: Performance comparison with prior work on standard VOS benchmarks. **Bold** indicates the best performance, and underline indicates the second-best performance.

| Method | $\mathcal{J}\&\mathcal{F}$ | | | | | $\mathcal{G}$ | $\mathcal{J}$ | $\mathcal{J}\&\dot{\mathcal{F}}$ |
|---|---|---|---|---|---|---|---|---|
| | SA-V val | SA-V test | LVOS v2 val | MOSE v1 val | DAVIS 2017 val | YTVOS 2019 val | M³-VOS core | MOSE v2 val |
| STCN (Cheng et al., 2021) | 61.0 | 62.5 | 60.6 | 52.5 | 85.4 | 82.7 | - | 29.7 |
| SwinB-AOT (Yang et al., 2021) | 51.1 | 50.3 | - | 59.4 | 85.4 | 84.5 | - | 30.2 |
| SwinB-DeAOT (Yang & Yang, 2022) | 61.4 | 61.8 | 63.9 | 59.9 | 86.2 | 86.1 | 62.3 | 32.6 |
| RDE (Li et al., 2022) | 51.8 | 53.9 | 62.2 | 46.8 | 84.2 | 81.9 | - | 32.0 |
| XMem (Cheng & Schwing, 2022) | 60.1 | 62.3 | 64.5 | 59.6 | 86.0 | 85.6 | 60.6 | 36.3 |
| SimVOS-B (Wu et al., 2023) | 44.2 | 44.1 | - | - | 88.0 | 84.2 | - | - |
| DEVA (Cheng et al., 2023) | 55.4 | 56.2 | - | 66.0 | 87.0 | 85.4 | - | 38.3 |
| ISVOS (Wang et al., 2023) | - | - | - | - | 88.2 | 86.3 | - | - |
| TarVIS (Athar et al., 2023) | - | - | - | - | 85.2 | - | - | - |
| UNINEXT (Yan et al., 2023) | - | - | - | - | 81.8 | 78.6 | - | - |
| UniVS (Li et al., 2024) | - | - | - | - | 76.2 | 71.5 | - | - |
| JointFormer (Zhang et al., 2025b) | - | - | - | - | 90.1 | 87.4 | - | 37.7 |
| Cutie-base (Cheng et al., 2024) | 60.7 | 62.7 | - | 69.9 | 87.9 | 87.0 | 64.6 | 42.8 |
| Cutie-base+ (Cheng et al., 2024) | 61.3 | 62.8 | - | 71.7 | 88.1 | 87.5 | - | - |
| SAM 2.1 (Ravi et al., 2025) | 78.6 | 79.6 | 84.1 | 74.5 | 90.6 | **88.7** | 64.9 | 49.5 |
| SAMURAI (Yang et al., 2024) | 79.8 | 80.0 | 84.2 | 72.6 | 89.9 | 88.3 | - | 51.1 |
| SAM2.1Long (Ding et al., 2025d) | 81.1 | 81.2 | 85.9 | 75.2 | **91.4** | 88.7 | 65.5 | 51.5 |
| **SeC (Ours)** | **82.7** | **81.7** | **86.5** | **75.3** | 91.3 | 88.6 | **67.2** | **53.8** |

Table 6: Ablation studies on proposed modules.

| Pixel-level Association | Concept Guidance | SA-V $\mathcal{J}\&\mathcal{F}$ | SeCVOS $\mathcal{J}\&\mathcal{F}$ |
|---|---|---|---|
| ✗ | ✗ | 78.6 | 58.2 |
| ✓ | ✗ | 82.4 | 62.2 |
| ✓ | ✓ | **82.7** | **70.0** |

Table 7: Ablation studies on LVLM size.

| LVLM Size | $\mathcal{J}\&\mathcal{F}$ | $\mathcal{J}$ | $\mathcal{F}$ |
|---|---|---|---|
| 1B | 68.4 | 68.2 | 68.7 |
| 2B | 69.5 | 69.3 | 69.8 |
| 4B | 70.0 | 69.7 | 70.2 |
| 8B | 70.3 | 70.1 | 70.7 |

## 5.3 COMPARISON ON STANDARD VOS BENCHMARKS

We further compare SeC against state-of-the-art methods on standard video object segmentation (VOS) benchmarks. The comparison encompasses both traditional matching-based segmentation algorithms and recent SAM 2 and its variants. As reported in Table 5, SeC achieves competitive or superior performance across all benchmarks. Specifically, it achieves leading $\mathcal{J}\&\mathcal{F}$ scores of 82.7 and 81.7 on the SA-V validation and test sets, and reaches 86.5 on LVOS v2. Furthermore, SeC significantly outperforms prior work on MOSE v2 with a $\mathcal{J}\&\dot{\mathcal{F}}$ score of 53.8 and on the M³-VOS core set with a $\mathcal{J}$ score of 67.2. These results establish SeC as the new state-of-the-art across a wide range of benchmarks, validating both the effectiveness and the versatility of our framework.

## 5.4 ABLATION STUDY

We conduct a series of ablation studies on the SA-V validation set and our proposed SeCVOS benchmark, with results presented in Table 6 and Table 7. Further experiments on our framework design and robustness are detailed in the Appendix C.1.

**Effectiveness of proposed modules.** Table 6 presents an ablation study evaluating the contributions of the pixel-level association and concept guidance modules. Enabling only the pixel-level association leads to a significant improvement on the SA-V benchmark and a modest gain on SeCVOS, highlighting its effectiveness in capturing low-level visual patterns, particularly beneficial in the single-shot scenarios of SA-V.

When the concept guidance module is further introduced, performance on SeCVOS improves by 7.8 points, demonstrating that concept-level reasoning is critical for handling the complex, multi-shot nature of SeCVOS, where simple pixel-level matching is insufficient. The marginal improvement on SA-V is expected, as this benchmark does not involve substantial semantic discontinuities or scene transitions that require high-level reasoning.

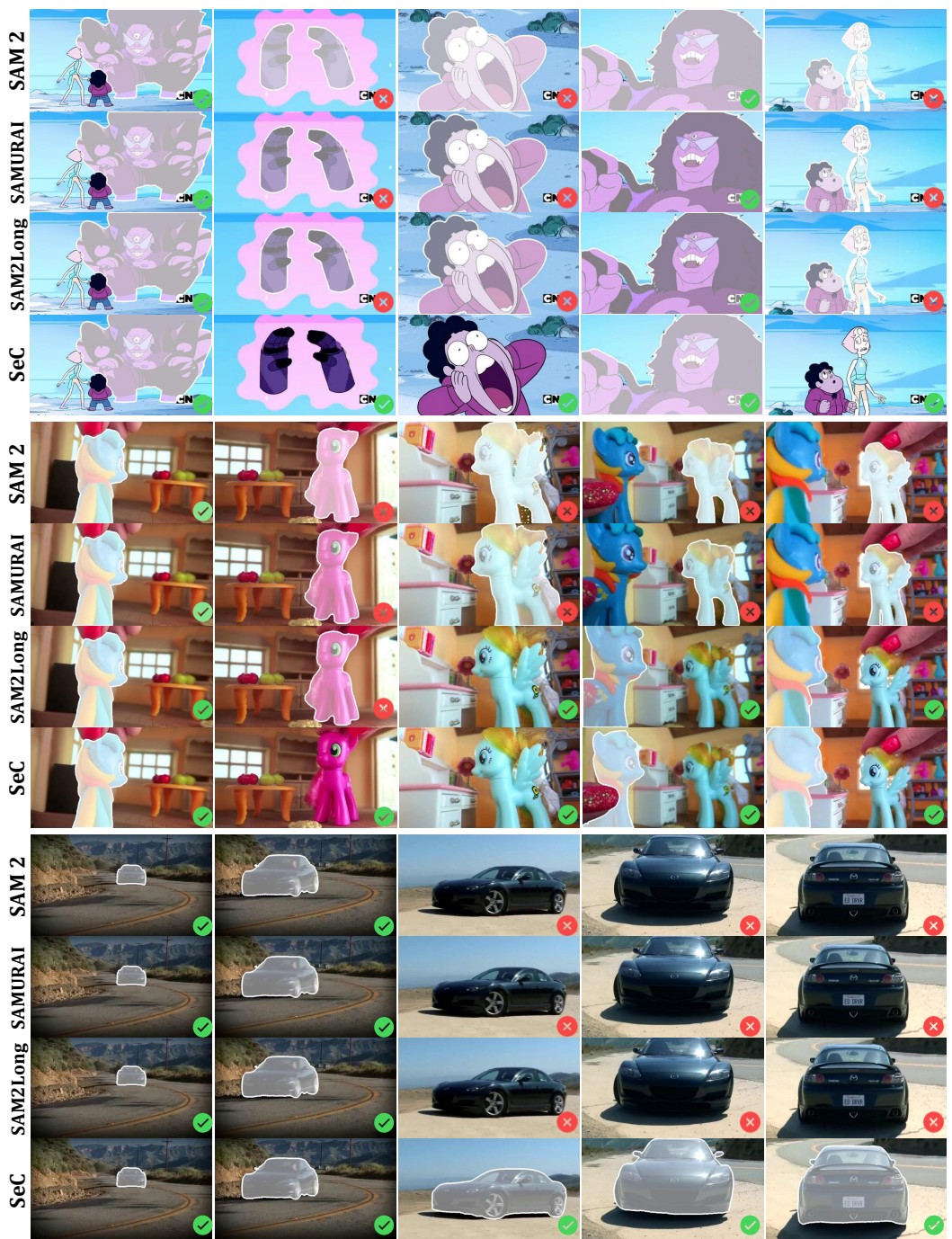

Figure 4: Qualitative comparison between SAM 2 (Ravi et al., 2025), SAMURAI (Yang et al., 2024), SAM2Long (Ding et al., 2025d) and SeC (ours) on the SeCVOS benchmark.

**Effectiveness of large vision-language model size.** Table 7 analyzes the effect of varying model parameter scales. As the parameter count increases from 1B to 4B, the model performance consistently improves across the three main metrics on the SeCVOS benchmark. However, further scaling to 8B leads to marginal gains, with results nearly identical to those of the 4B model. This indicates that beyond a certain scale, the benefits of increasing model size begin to saturate, and no longer translate into proportional performance improvements.

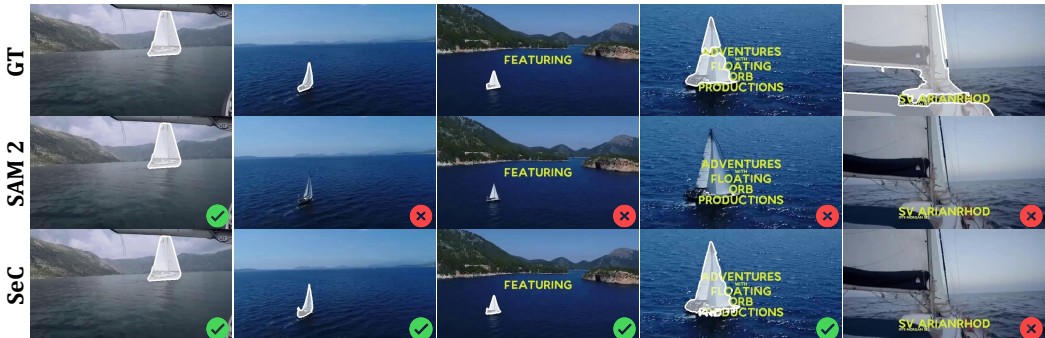

Figure 5: Failure case example from the SeCVOS benchmark.

## 5.5 VISUALIZATION

To more intuitively demonstrate the segmentation performance of our method, Figure 4 showcases a visual comparison between our approach and the SAM 2 baseline on the SeCVOS benchmark. Compared to the SAM 2, our SeC model consistently delivers reliable segmentation results by constructing a well-formed concept representation, particularly in handling complex situations such as viewpoint changes, background interference, and object occlusion. More qualitative results on the SeCVOS and MOSE v2 benchmark can be found in the Appendix C.2.

However, since this concept is learned from a limited set of viewpoints, the model's performance may decline in extreme cases where the current viewpoint differs significantly from those encountered during concept construction. For example, as illustrated in Figure 5, the interior view of the sailboat in the fifth image poses a challenge. The drastic shift in perspective leads to a failure in matching the frame to the learned concept, resulting in incorrect segmentation.

## 6 CONCLUSION

We present Segment Concept (SeC), a novel concept-driven framework for Semi-Supervised Video Object Segmentation that moves beyond traditional appearance-based matching by leveraging high-level object-centric reasoning. By integrating the conceptual perception capabilities of Large Vision-Language Models (LVLMs), SeC constructs and updates robust semantic representations over time, enabling consistent tracking under challenging conditions such as dynamic scene or appearance transitions. To evaluate these capabilities, we introduce SeCVOS, a new benchmark specifically designed to test semantic-level understanding in complex, multi-shot video scenarios. Extensive experiments show that SeC significantly outperforms existing state-of-the-art models, including SAM 2 and its variants, across both SeCVOS and standard benchmarks, while maintaining competitive efficiency. We hope SeC and SeCVOS will inspire further exploration of concept-level modeling for long-term and semantically grounded video understanding.

Interestingly, recent SAM 3 (Carion et al., 2025) also highlights segmentation with concepts, which shares a highly similar core idea with our work in emphasizing the importance of semantics. However, their focuses differ. Our SeC focuses more specifically on complex video scenarios, whereas SAM 3 is more of a system-level work that empowers image grounding and does not substantially modify the video segmentation architecture as SAM 2 did. Building on this, a promising direction for future work is to introduce concept-level binding directly into video scenarios, particularly for handling prompts with strong temporal relationships, such as "a child who is running."

## ACKNOWLEDGMENTS

This project is funded in part by Shanghai Artificial Intelligence Laboratory, Shanghai Innovation Institute, the Centre for Perceptual and Interactive Intelligence (CPII) Ltd under the Innovation and Technology Commission (ITC)'s InnoHK. Dahua Lin is a PI of CPII under the InnoHK.

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

SUPPLEMENTARY MATERIALS

This supplementary material provides further details about our SeC framework and the SeCVOS benchmark, including the model's architecture and training, as well as the benchmark's composition and supported tasks. We also present additional ablation studies and qualitative results to further validate the effectiveness of our approach. Additionally, we discuss current limitations, ethical considerations and the intended scope of use for SeCVOS. A demo video and additional video cases from the benchmark are also provided in the zip. The appendix is organized as follows:

- Section A. SeC Training Details
- Section B. SeCVOS Benchmark Details
- Section C. Supplementary Experiment
- Section D. Evaluation Benchmark Details
- Section E. Limitations, Reproducibility, and Broader Impact
- Section F. LLM Usage

# A  SeC Training Details

We adopt a two-stage training approach: (1) training the pixel-level association memory module for long-term temporal modeling, and (2) fine-tuning the LVLM-based semantic guidance module for concept modeling.

In the first stage, we train the pixel-level association memory using 2k videos from the SA-V training set, selected based on the highest number of scene transitions as detected by SceneDetect. For each video, 24 shuffled frames are randomly sampled for training. During this stage, only the memory attention module is updated, while all other components remain frozen. The model is trained for 40 epochs with a batch size of 64 and a learning rate of $5 \times 10^{-6}$.

In the second stage, we fine-tune InternVL 2.5-4B (Chen et al., 2024d) on approximately 190k object instances from the SA-V training set, each containing at least three visible masks. For each training sample, 1 to 7 reference frames are randomly selected. Instead of overlaying an alpha-blended object mask, we draw a green contour around the target object. This contour effectively highlights the segmentation target without obstructing the visual features needed for LVLM-based perception. Among these, 0 to 2 are distractor frames containing incorrect annotations, while the rest provide valid visual prompts. Additionally, one non-overlapping query frame is included. All images are resized to $448 \times 448$ resolution. We apply LoRA-based fine-tuning to the InternVL 2.5, while keeping all SAM 2 parameters frozen. The model is trained for 3 epochs with a batch size of 64 and a learning rate of $4 \times 10^{-5}$.

All experiments are conducted on 8 NVIDIA A800 GPUs, and the loss function remains consistent with that of SAM 2.

# B  SeCVOS Benchmark Details

## B.1  Video Details

The SeCVOS benchmark comprises a diverse collection of video sequences designed to rigorously evaluate video object segmentation and tracking performance. The videos range from 6 to around 60 seconds in length and average 29.4 seconds. They span a variety of contexts including indoor, outdoor, and animated scenes, and feature targets such as humans, vehicles, and animals. The benchmark intentionally incorporates significant difficulties like abrupt scene transitions, rapid object motion, and severe environmental interference, most notably frequent occlusions and the intermittent appearance of targets. Each sequence is accompanied by meticulously reviewed, high-quality ground-truth masks that capture the precise shape and positional evolution of the target over time. Figure 6 provides examples of these annotations. The primary goal of SeCVOS is to advance the state-of-the-art by challenging existing methods with dynamic and complex visual conditions. We will release SeCVOS as an open-source benchmark to support future research in concept-driven video object segmentation.

**Time Flow**

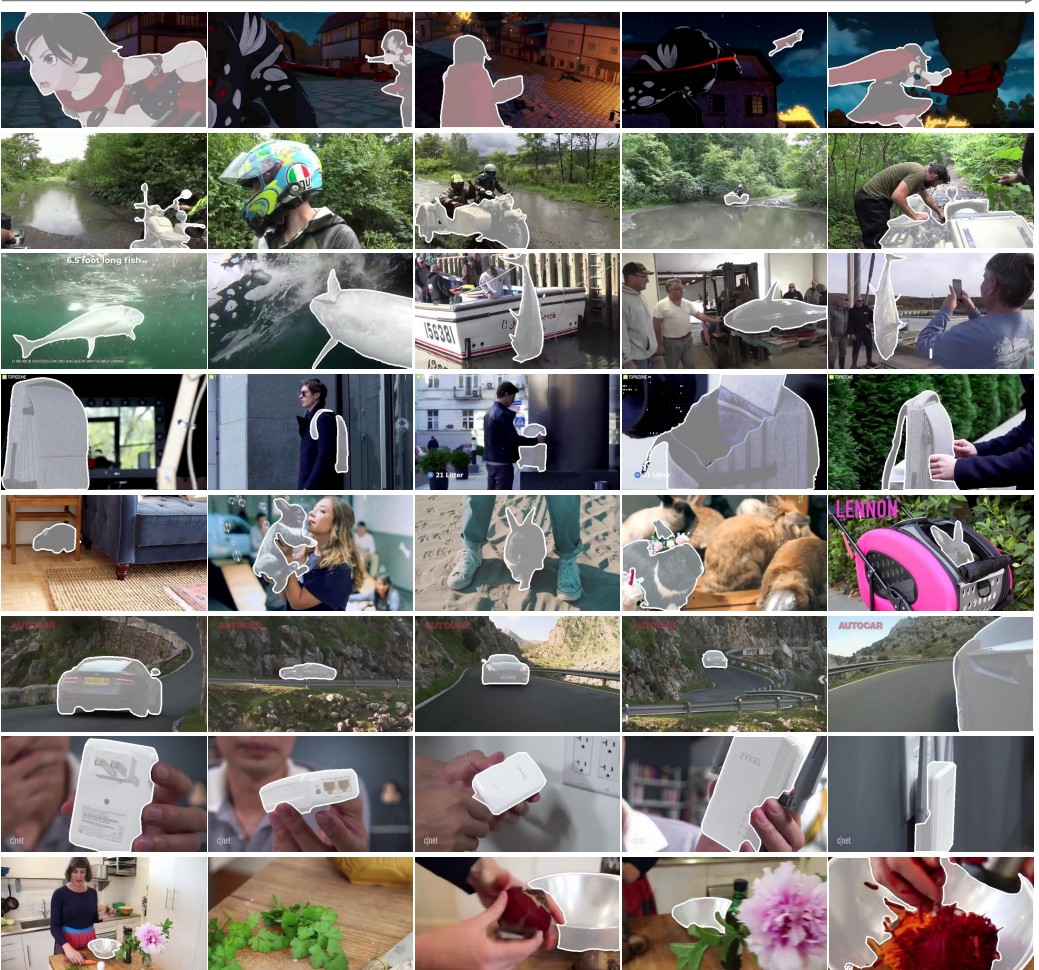

Figure 6: Example video sequences from the SeCVOS benchmark with overlaid target masks. Each row corresponds to frames from a single video sequence, illustrating the annotated object masks.

## B.2 REFERRING VIDEO OBJECT SEGMENTATION ON SECVOS

In addition to the semi-supervised Video Object Segmentation task, our proposed SeCVOS dataset also supports the Referring Video Object Segmentation task. In this task, we generate detailed descriptions for each object in the SeCVOS dataset. These descriptions are initially generated by the Gemini 2.5 Pro (Team et al., 2024) and subsequently refined through rigorous manual verification and editing to ensure accuracy. Figure 7 depicts several data samples, and notably, in the presence of visually similar distractor objects, we provide additional fine-grained descriptions to support precise model discrimination of the target objects.

Under this setting, we evaluated several state-of-the-art RefVOS methods, including both LVLM based approaches and traditional temporal propagation baselines. As shown in Table 8, the performance of all methods on the SeCVOS benchmark remains limited. VISA (Yan et al., 2024) and GLUS-A (Lin et al., 2025) performed comparatively better, possibly because they were trained on datasets with more complex textual instructions, which helps with

Table 8: Performance comparison on Ref-SeCVOS.

| Method | Total Params | Ref-SeCVOS | | |
|---|---|---|---|---|
| | | $\mathcal{J}$ | $\mathcal{F}$ | $\mathcal{J}\&\mathcal{F}$ |
| *Propagation Based Method* | | | | |
| Grounded SAM 2 (Ren et al., 2024) | 400 M | 48.4 | 49.3 | 48.9 |
| SAMWISE (Cuttano et al., 2025) | 210 M | 54.1 | 53.9 | 54.0 |
| *LVLM Based Method* | | | | |
| VideoLISA (Bai et al., 2024) | 3.8 B | 43.7 | 41.8 | 42.8 |
| Sa2VA (Yuan et al., 2025) | 8 B | 51.5 | 52.0 | 51.8 |
| GLUS-A (Lin et al., 2025) | 7 B | 59.7 | 60.0 | 59.8 |
| VISA (Yan et al., 2024) | 7 B | **60.4** | **58.6** | **59.5** |

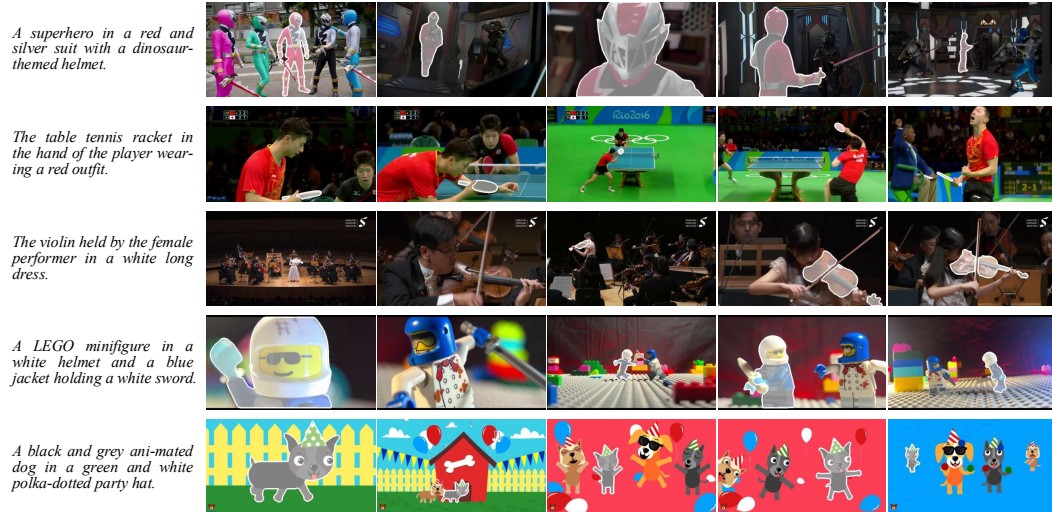

| | | | | | |
|---|---|---|---|---|---|
| *A superhero in a red and silver suit with a dinosaur-themed helmet.* | | | | | |
| *The table tennis racket in the hand of the player wearing a red outfit.* | | | | | |
| *The violin held by the female performer in a white long dress.* | | | | | |
| *A LEGO minifigure in a white helmet and a blue jacket holding a white sword.* | | | | | |
| *A black and grey animated dog in a green and white polka-dotted party hat.* | | | | | |

Figure 7: Example video sequences and corresponding referring expressions from the SeCVOS.

cross-modal reasoning and object discrimination. Overall, these results highlight the challenges of the SeCVOS benchmark in terms of scene complexity, fine-grained language descriptions, and visual discrimination, indicating that there is still significant room for improvement in RefVOS.

## C  SUPPLEMENTARY EXPERIMENT

### C.1  ADDITIONAL ABLATION STUDIES

In this section, we conduct further ablation studies to validate the design choices and robustness of our framework. We specifically investigate three aspects: the impact of the number of concept tokens, the framework's independence from the choice of scene detector, and its resilience to noisy guidance from the LVLM.

**Ablation on the number of concept tokens.** To determine the ideal representational capacity, we investigated the effect of varying the number of concept tokens. As shown in Table 9, the results clearly indicate that increasing the token count from one to four yields no significant performance gains across all metrics. This outcome validates our use of a single, dense token embedding as a more efficient and equally effective approach compared to a multi-token representation for this task.

**Robustness to Scene Detector Choice.** To further validate our framework's robustness with different scene detectors, we conducted a supplementary experiment comparing several different lightweight scene-change detection algorithms on SeCVOS. We compared our HSV-based method against techniques based on pixel-wise absolute difference (ABS DIFF), structural similarity (SSIM), optical flow (FLOW), and feature matching with Oriented FAST and Rotated BRIEF (ORB). As demonstrated in Table 10, our framework exhibits robustness to the choice of scene detector and performs competitively across all of them.

Table 9: Performance comparison of different number of concept tokens on SeCVOS.

| #Concept Tokens | $\mathcal{J}\&\mathcal{F}$ | $\mathcal{J}$ | $\mathcal{F}$ |
|---|---|---|---|
| 1 | 70.0 | 69.7 | 70.2 |
| 2 | 70.0 | 69.7 | 70.3 |
| 4 | 69.9 | 69.6 | 70.2 |

Table 10: Performance comparison of different scene detectors on SeCVOS.

| Scene Detector | $\mathcal{J}\&\mathcal{F}$ | $\mathcal{J}$ | $\mathcal{F}$ |
|---|---|---|---|
| ABS DIFF | 69.0 | 68.8 | 69.3 |
| ORB | 68.7 | 68.5 | 69.0 |
| SSIM | 69.3 | 69.1 | 69.6 |
| FLOW | 69.5 | 69.3 | 69.7 |
| HSV (ours) | 70.0 | 69.7 | 70.2 |

**Robustness to Noisy LVLM Guidance.** To assess the system's resilience against unreliable outputs from the LVLM, we performed an additional experiment on SeCVOS. In this experiment, we forced LVLM intervention on every frame while intentionally introducing a varying number of incorrect masks($n$) as input noise. As shown in Figure 8, the system's performance degrades gracefully and consistently outperforms baselines as noise increases, rather than failing abruptly. A significant drop was observed only when the number of noisy frames exceeded the number of clean ones (*i.e.*, when $n > 4$). This resilience is attributed to two core design principles: 1) our framework fuses LVLM guidance with visual features rather than replacing them, which mitigates the impact of any single inaccurate mask; and 2) the LVLM was explicitly trained with noisy inputs, inherently improving its robustness.

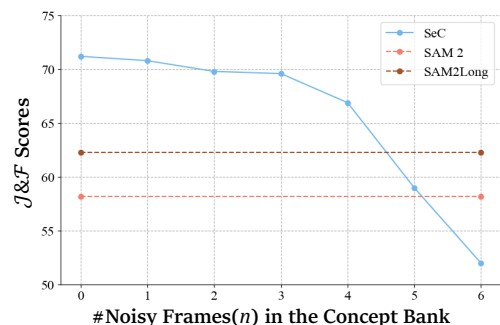

Figure 8: $\mathcal{J}\&\mathcal{F}$ Curve in terms of the number of noisy masks on SeCVOS. SeC degrades gracefully and consistently outperforms baselines.

### C.2 ADDITIONAL QUALITATIVE RESULTS

To further illustrate our method's capabilities, we present additional qualitative results for the SeCVOS and MOSE v2 benchmarks in Figure 9 and Figure 10, respectively.

We also provide some additional qualitative examples in Figure 12 to show SeC's robustness in real applications. SeC maintains stable tracking in crowded scenes with heavy occlusions and re-appearances, consistently follows distant vehicles under illumination changes in dashcam videos, and accurately distinguishes targets from distractors in wildlife footage. Together, these results confirm that SeC delivers more robust and reliable segmentations than pixel-matching-based methods under a variety of challenging conditions.

These visualizations demonstrate the robustness and effectiveness of our SeC in scenarios that closely approximate real-world conditions. Our method delivers consistently reliable tracking and segmentation, even within challenging videos featuring drastic appearance and scene variations.

### D EVALUATION BENCHMARK DETAILS

To evaluate our method, we select seven standard VOS benchmarks. Following established evaluation protocols, we report the standard VOS metrics of region similarity ($\mathcal{J}$), contour accuracy ($\mathcal{F}$), and their average ($\mathcal{J}\&\mathcal{F}$) for SA-V, LVOS v2, MOSE v1 and DAVIS. For MOSE v2, we report the improved $\mathcal{J}\&\dot{\mathcal{F}}$ score proposed by Ding et al. (2025a), and for M³-VOS, we report $\mathcal{J}$. The benchmarks used for evaluation are detailed as follows:

**SA-V** (Ravi et al., 2025) is a large-scale dataset for promptable video segmentation, containing over 50.9K video clips and 35.5M annotated masks. The dataset is divided into training, validation, and testing sets, with 155 videos for validation and 150 for testing. Its core challenge lies in segmenting small, occluded, and reappearing objects across diverse scenarios.

**LVOS v2** (Hong et al., 2024) expands upon LVOS v1 for long-term video object segmentation. It is split into 420 videos for training, 140 for validation, and 160 for testing. The dataset includes 44 categories, with 12 of these deliberately held out from the training set to specifically evaluate the generalization capabilities of VOS models.

**DAVIS 2017** (Pont-Tuset et al., 2017) is a foundational benchmark that established the standard for multi-object video segmentation, advancing from its single-object predecessor. It consists of 150 sequences, which are divided into 60 for training, 30 for validation, and 60 for testing. The dataset increases complexity with challenges like severe occlusions and fast motion.

**YTVOS 2019** (Xu et al., 2018) is a benchmark designed to evaluate a model's ability to generalize to unseen object categories. The 2019 version contains over 4,500 videos, with its validation and test sets including a mix of 65 "seen" categories from training and dozens of "unseen" categories.

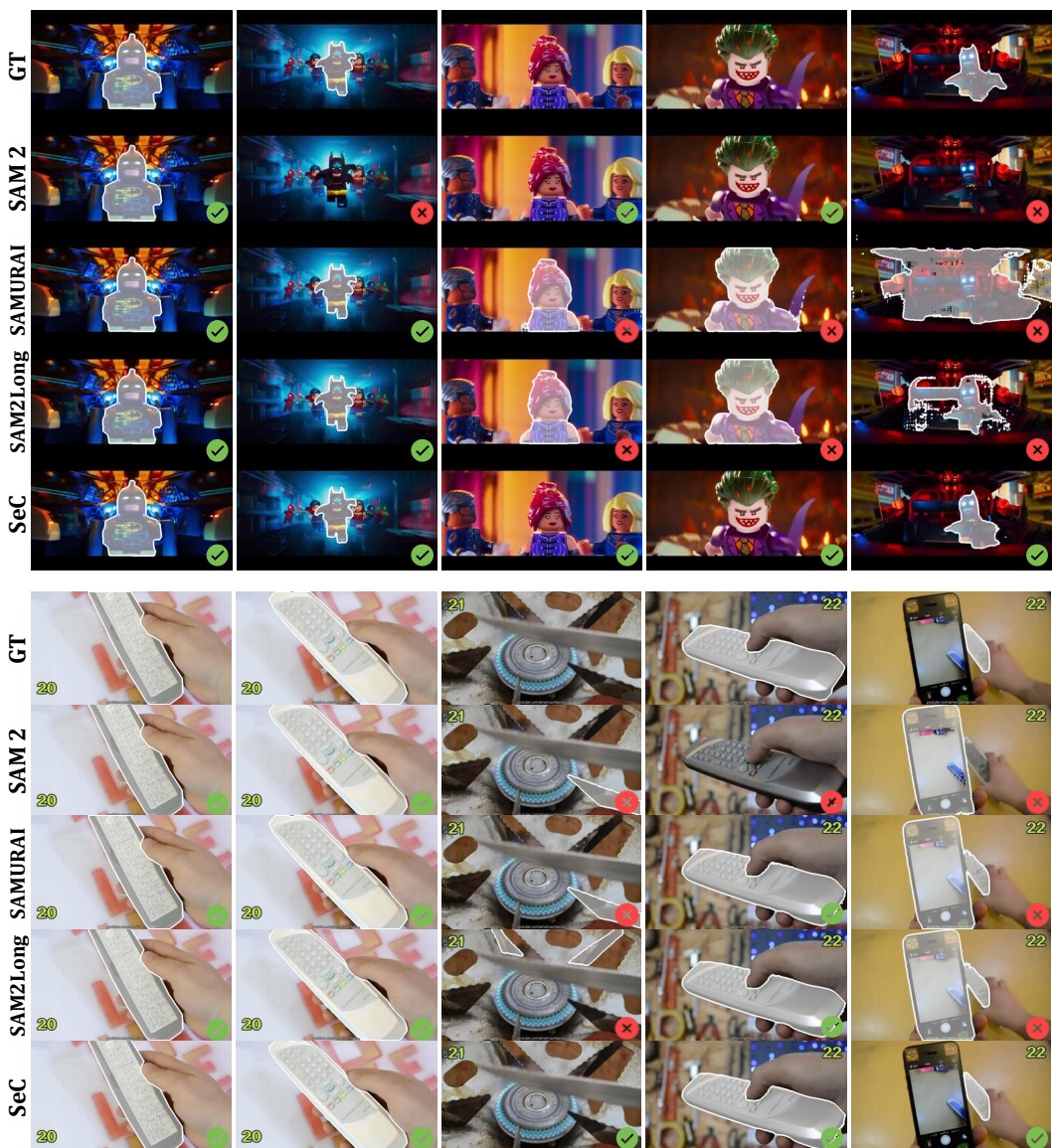

Figure 9: Additional qualitative comparison between SAM 2 (Ravi et al., 2025), SAMURAI (Yang et al., 2024), SAM2Long (Ding et al., 2025d) and SeC (ours) on the SeCVOS, with GT (Ground Truth) provided for reference.

**MOSE v1** (Ding et al., 2023) is a large-scale benchmark created to evaluate VOS methods in complex, realistic scenarios where objects are not always salient or isolated. It contains 2,149 video clips with over 431,000 high-quality masks for 5,200 objects across 36 categories, which are divided into 1,507 videos for training, 311 for validation, and 331 for testing. The dataset is specifically designed to include challenging situations such as heavy object occlusion, crowded scenes, and frequent disappearance and reappearance of targets.

**MOSE v2** (Ding et al., 2025a) builds upon MOSE v1 to expose the limitations of state-of-the-art VOS methods in complex, real-world scenarios. It consists of 5,024 videos and over 701,976 highquality masks for 10,074 objects across 200 categories, which are split into 3,666 for training, 433 for validation, and 614 for testing. Compared to its predecessor, MOSE v2 introduces novel adversarial conditions such as adverse weather, low-light scenes, camouflaged objects and nonphysical targets like shadows and reflections.

**M³-VOS** (Chen et al., 2025) introduces the novel challenge of segmenting objects that undergo significant morphological and appearance changes due to phase transitions (*e.g.*, melting, dissolving,

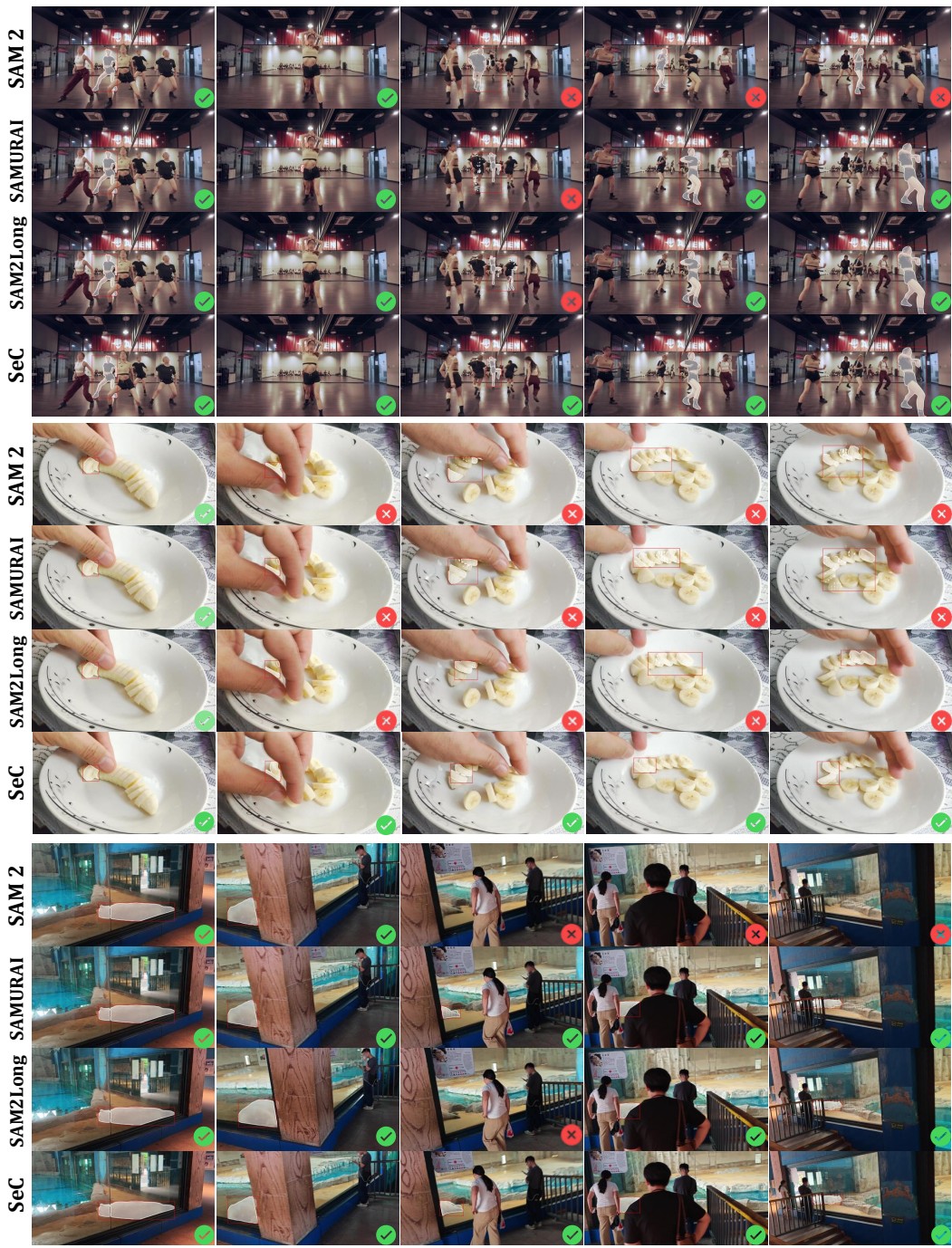

Figure 10: Additional qualitative comparison between SAM 2 (Ravi et al., 2025), SAMURAI (Yang et al., 2024), SAM2Long (Ding et al., 2025d) and SeC (ours) on the MOSE v2(Ding et al., 2025a).

flowing). Comprising 479 high-resolution videos, this benchmark directly challenges the core assumption of appearance consistency that underpins many VOS models by focusing on the physical dynamics of objects.

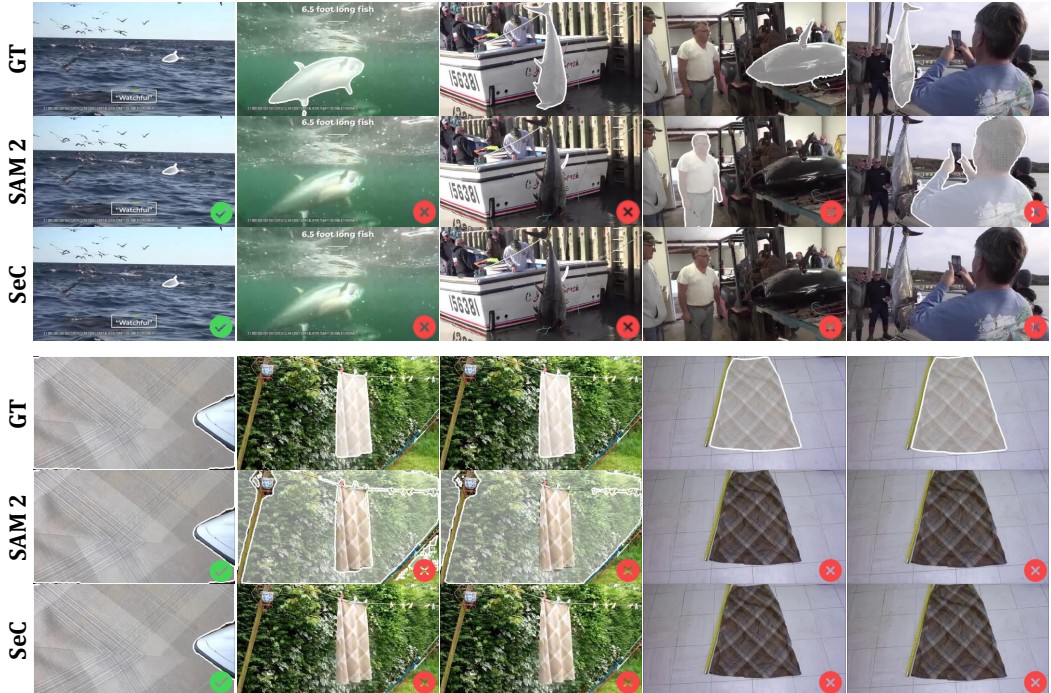

Figure 11: Additional failure case example from the SeCVOS benchmark.

# E  LIMITATIONS, REPRODUCIBILITY, AND BROADER IMPACT

## E.1  LIMITATIONS

Despite its promising results, our work still leaves room for improvement. First, the current transition detection mechanism is lightweight and simple, but may fail in certain edge cases. As shown in the figure 11, SeC struggles when the encountered viewpoint deviates significantly from those observed during concept construction, or when the LVLM lacks sufficient prior knowledge to form a reliable concept of the target. A more robust approach would involve learning a dynamic indicator to decide when to invoke LVLM-based reasoning and better tolerate viewpoint variation. Second, although SeCVOS introduces multi-shot complexity, its overall video length remains shorter than that of existing datasets like LVOS (Hong et al., 2024). While SeCVOS already presents significant challenges for current methods, extending it with longer-duration videos would further evaluate the temporal reasoning capabilities of future models.

## E.2  REPRODUCIBILITY

To ensure the reproducibility of our work and contribute to the broader academic community, we provide comprehensive details of our proposed Segment Concept (SeC) framework and experimental implementation in Section 3 and 5.1. Our framework are built upon the publicly available models (InternVL 2.5 and SAM 2) and experiments were conducted on standard public datasets (such as SA-V, LVOS, and MOSE). In line with this commitment, we will release our Semantic Complex Scenarios Video Object Segmentation (SeCVOS) benchmark, model checkpoints, and the complete source code for training and inference. We hope these resources will serve as a valuable reference for future VOS applications, fostering innovation and accelerating progress within the field.

## E.3  BROADER IMPACT

The SeCVOS benchmark is constructed using only publicly available video data, which is used exclusively for academic research purposes. All annotation work was performed by volunteers who

were fully informed about the nature of the project. No private, sensitive, or restricted data were used.

The goal of our research is to support the development of technologies that can positively impact society, such as autonomous systems, assistive technologies, and tools for enhanced human-computer interaction. However, we acknowledge the potential risks associated with the misuse of segmentation technologies, including privacy concerns and unauthorized surveillance. We encourage the responsible use of our benchmark and methods, and explicitly discourage any applications that may infringe upon personal privacy or be deployed for harmful purposes.

All annotations and experimental results presented in this work were generated solely for research purposes and adhere to ethical guidelines regarding the use of public visual data within the academic community.

## F   LLM USAGE

During the writing process, we utilized Large Language Models (LLMs) as a tool to aid in editing and polishing the language. The core ideas, analysis, and conclusions presented in this paper are the work of the authors. The LLMs were used solely to improve grammar, clarity, and readability.

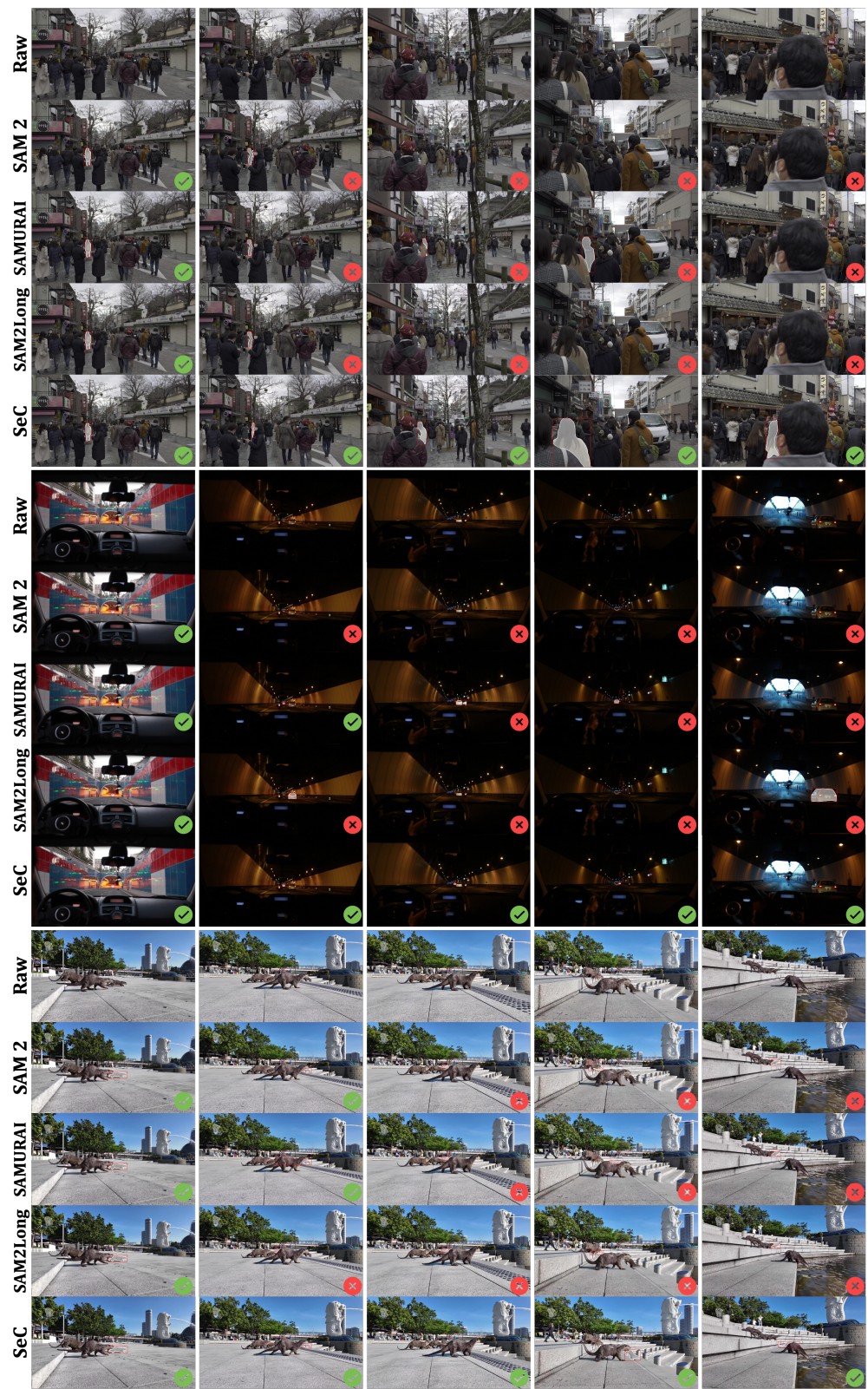

Figure 12: Additional qualitative comparison between SAM 2 (Ravi et al., 2025), SAMURAI (Yang et al., 2024), SAM2Long (Ding et al., 2025d) and SeC (ours) on scenes with complex motion, occlusion, or lighting variations, with the raw image provided for reference.

