# OpenReview forum: "Advancing Complex Video Object Segmentation via Progressive Concept Construction"
_ICLR.cc/2026/Conference — ICLR 2026 Poster_

### Official Review · Reviewer_V92j · 2025-10-30

**Soundness:** 2
**Presentation:** 3
**Contribution:** 2
**Rating:** 4
**Confidence:** 4

**Summary:**

This paper proposes Segment Concept (SeC), a concept-driven framework for video object segmentation (VOS) that shifts from traditional appearance-based matching to progressive construction of object-level concepts using Large Vision-Language Models (LVLMs). The method selectively activates LVLM-based concept reasoning only when significant scene changes are detected, balancing semantic understanding with computational efficiency. Additionally, the authors introduce SeCVOS, a new benchmark designed to evaluate VOS models in semantically complex, multi-shot scenarios. Experiments show that SeC outperforms state-of-the-art methods, including SAM 2 and its variants, on both SeCVOS and standard VOS benchmarks.

**Strengths:**

1. Benchmark Contribution (SeCVOS ).
2. Rigorous Evaluation and Ablation. The paper includes extensive experiments, ablation studies, and qualitative analyses. The comparison with a wide range of methods and the inclusion of failure cases (Fig. 5) enhance credibility.
3. Good performance. The results are consistent across multiple datasets and settings, demonstrating robustness and generalizability.

**Weaknesses:**

1. Limited Comparison with LVLM-Based VOS Methods:
While the paper compares with many traditional and SAM-based methods, it lacks a thorough comparison with recent LVLM-based VOS approaches (e.g., VISA, GLUS, VideoLISA). Table 8 in the appendix is a start, but it should be integrated into the main paper to better position SeC in the LVLM-VOS landscape.
2. Scene Change Detection Simplicity:
The scene change detector is based on a simple HSV histogram difference. This may not be robust to semantically significant but visually subtle changes. A learning-based or more sophisticated detector could be more reliable.
3. Computational Overhead:
Although the authors emphasize efficiency, the use of LVLMs (even sparsely) still introduces significant computational cost. A more detailed analysis of latency, memory usage, and comparison with other LVLM-based methods would be helpful.
4. Lack of User Study or Qualitative Justification for "Concept":
The term "concept" is central to the paper but remains somewhat abstract. A user study or qualitative analysis showing how the learned concept tokens align with human-understandable semantics would strengthen the claim.
5. Lack of User Study or Qualitative Justification for "Concept":
The term "concept" is central to the paper but remains somewhat abstract. A user study or qualitative analysis showing how the learned concept tokens align with human-understandable semantics would strengthen the claim.

**Questions:**

See the weaknesses.

---

> ### Author Response · Authors · 2025-11-21
> **Response to Reviewer V92j (1/2)**
>
> We sincerely thank you for your constructive comments and meticulous review, especially for acknowledging that:
>
> - Contribution of SeCVOS Benchmark
> - Rigorous Evaluation and Ablation
> - Robustness and generalizability of our framework
>
> We have conducted additional experiments and will revise our manuscript accordingly. Our detailed responses are below.
>
> > **W1: Lack of comparison with recent LVLM-based VOS methods (e.g., VISA, GLUS, VideoLISA).**
>
> Thank you for the constructive suggestion. We have added the comparison table to the main paper (Table 9). We would like to clarify that SeC differs from recent LVLM-VOS approaches (VISA, GLUS, VideoLISA), which focus on referring VOS and rely on textual prompts. To the best of our knowledge, SeC is among the first attempts to bring LVLM reasoning into Semi-VOS, analyzing visual cues across the video to construct a concept of the target, enabling robust handling of drastic appearance and scene variations, without any textual input.
>
> In addition, we also provide a text-supported variant, Grounded SeC. Similar to Grounded SAM2, we employ the open-vocabulary detector Grounding DINO to obtain the first-frame mask, and then use SeC to propagate the predictions throughout the video. Experiments show that Grounded SeC significantly outperforms existing LVLM-VOS models in both accuracy and efficiency, though its performance is still lower than vanilla SeC, which is unsurprising given that detector-generated masks are inherently weaker than the ground-truth initialization used in Semi-VOS.
>
> | Input | Method        | FPS  | Params | Mem(GiB) |   J   |   F  |  J&F |
> | :---: | ------------- | ---: | -----: | :------: | :---: | :--: | :--: |
> | Text  | VideoLISA     | 1.4  | 3.8B   | 12.8     | 43.7  | 41.8 | 42.8 |
> | Text  | Sa2VA         | 10.2 | 8B     | 18.3     | 51.5  | 52.0 | 51.8 |
> | Text  | GLUS-A        | 6.7  | 7B     | 14.7     | 59.7  | 60.0 | 59.8 |
> | Text  | VISA          | 4.3  | 7B     | 17.1     | 60.4  | 58.6 | 59.5 |
> | Text  | Grounded SeC  | 14.8 | 4B     | 11.4     | 62.6  | 63.3 | 62.9 |
> | Mask  | SeC           | 14.8 | 4B     | 11.4     | 69.7  | 70.2 | 70.0 |
>
> > **W2: The HSV-based scene change detector is simple and may not be robust to semantically significant but visually subtle changes. A learning-based or more sophisticated detector could be more reliable.**
>
> Thank you for the insightful comment. We acknowledge that our HSV-based scene detector is relatively simple but it is effective in practice. Our core philosophy is that the efficient pixel-level memory should handle gradual changes, while the computationally intensive LVLMs should only be activated for drastic scene shifts where high-level reasoning is necessary. Our ablation study (Figure 3) confirms that activating the LVLM on fewer than 10% of frames yields most of the performance gains.
>
> Moreover, our choice of a simple detector is efficiency-driven, and SeC is in fact insensitive to the detector design. As shown in Table 10 of the appendix, replacing HSV with several other lightweight detectors achieves comparable performance, indicating strong robustness.
>
> We agree that stronger detectors may yield further improvements. Accordingly, we conducted an additional experiment that adopts InternVL3-38B as the scene detector, which leads to additional performance gains. This result suggests that learning-based or more sophisticated detectors are a promising direction for future work.
>
> | Model               | J&F   |  J    |   F   |
> |---------------------|-------|-------|-------|
> | HSV-based detector  | 70.0  | 69.7  | 70.2  |
> | InternVL as detector| 71.3  | 71.1  | 71.5  |
>
> > **W3: Computational Overhead. A more detailed analysis of latency, memory usage, and comparison with other LVLM-based methods would be helpful.**
>
> Thank you for the constructive suggestion. Although SeC integrates LVLM reasoning, it remains significantly faster and more memory-efficient than existing LVLM-VOS methods, because (1) LVLMs are activated sparsely, and (2) SeC uses only a single forward pass without autoregressive decoding.
>
> As shown in the table provided in our response to W1, we have included a more comprehensive analysis of latency and memory usage, along with comparisons to existing LVLM-VOS methods. The results demonstrate that SeC achieves significantly better speed and lower memory consumption than other LVLM-based VOS approaches, while also delivering better accuracy.

---

> ### Author Response · Authors · 2025-11-21
> **Response to Reviewer V92j (2/2)**
>
> > **W4 & W5: Lack of User Study or Qualitative Justification for "Concept": The term "concept" is central to the paper but remains somewhat abstract.**
>
> Thank you for pointing out the need for a more concrete justification of "concept." To further validate that our concept tokens indeed align with human-understandable semantics, we conducted a user study and a concept-retrieval experiment.
>
> **User Study.**
>
> We carried out a user study with 17 participants on 10 videos. For each video, we provided a brief text description and presented anonymized segmentation results from four methods (SAM2, SAMURAI, SAM2Long, and ours SeC). Participants were asked to rate each result along two axes: (1) temporal consistency across frames, and (2) semantic alignment with the textual description, using a 1–5 Likert scale.
>
> As shown in the table below, our method achieved the highest scores on both axes, indicating that human subjects consistently found our concept-guided segmentation to be both more stable across frames and more semantically aligned with the textual prompt.
>
> | Method    | Consistency   | Semantic Alignment   |
> | --------- | ------------- | -------------------- |
> | **Ours**  | **4.52**      | **4.63**             |
> | SAM2Long  | 3.45          | 3.41                 |
> | SAMURAI   | 3.27          | 3.10                 |
> | SAM2      | 3.05          | 2.96                 |
>
>
> **Retrieval Experiment.**
>
> To further evaluate whether the learned concept tokens capture intrinsic object semantics, we performed a retrieval experiment on a subset of SeCVOS. For each video, we generated two concept embeddings: one from the first half and one from the second. The task was to use the second-half embedding to retrieve the corresponding first-half embedding from a set of candidates. To ensure that the evaluation focuses on concept-level semantics rather than contextual cues, we constructed a challenging "distractor" setting by replacing the backgrounds of the second-half frames with random scenes. We then compared our concept embedding with the SAM2 object pointer under this setting.
>
> As shown in the table below, our concept embedding substantially outperforms the baseline, especially under the distractor condition.
>
> | Method                  | Original Acc@1 | Original Acc@3 | Distractor Acc@1  | Distractor Acc@3  | Overall Acc@1  | Overall Acc@3  |
> |-------------------------|----------------|----------------|-------------------|-------------------|----------------|----------------|
> | Object Pointer (SAM 2)  | 30.0%          | 55.0%          | 15.7%             | 37.9%             | 17.5%          | 40.0%          |
> | Concept Embedding (Ours)| 80.0%          | 90.0%          | 55.7%             | 73.4%             | 58.8%          | 75.6%          |
>
> Together, these results provide qualitative and quantitative evidence that our concept representation captures robust, human-understandable object-level semantics.
>
> ---
> Thank you again for the constructive comments, which have helped us significantly improve the paper's quality! Please don't hesitate to let us know if there are any additional clarifications or experiments that we can offer!

---

### Official Review · Reviewer_tKVK · 2025-10-30

**Soundness:** 3
**Presentation:** 3
**Contribution:** 2
**Rating:** 4
**Confidence:** 5

**Summary:**

This paper presents a SAM2 based methods for long term video object segmentation. To handle the underlying occusion and  object re-detection, the authors employ LVLMs to extract the high-level semantic information about the  segmentation target and realize the consistent tracking.  The authors evaluate the proposed method on several public VOS datasets as well as the self-collected SeCVOS dataset.

**Strengths:**

The statement is clear, and the figures are in good illustration.

**Weaknesses:**

1. Novelty is a big issue. Token-level video summarization has been widely exploited for
long-term video understanding tasks, such as [1,2,3].
[1]Online Video Understanding: A Comprehensive Benchmark and Memory-Augmented Method
[2] InternVideo2.5: Empowering Video MLLMs with Long and Rich Context Modeling.
[3]  Streaming Long Video Understanding with Large Language Models, neurips

2. Fairness about the selected dataset called SeCVOS benchmark. This dataset is small with only 160 manually videos.
Also, compared to the existing vos dataset LVOS, the duration is shorter than LVOS. I think the authors can test the proposed methods on large-scale tracking datasets, such as TrackingNet and SportsMOT.


3. In the experimental section, the compared methods are all before 2025. Many recent counterparts are not compared. Also, the performance advantage over these old methods is slight.

4. The method section is too simple and straightforward.


Overall, the contribution of the whole paper is weak.

**Questions:**

The computation burden should be analyzed.

---

> ### Author Response · Authors · 2025-11-21
> **Response to Reviewer tKVK (1/2)**
>
> Thank you for your valuable time and feedback on our manuscript. We hope the following clarifications and explanations can resolve your concerns and help you better appreciate the contribution of our work.
>
> > **W1: Novelty is a big issue. Token-level video summarization has been widely exploited for long-term video understanding tasks, such as [1,2,3].**
>
> We thank the reviewer for the feedback and would like to clarify the core novelty of SeC is not video token summarization, but a fundamentally different modeling perspective.
>
> 1. **The tasks differ fundamentally.** The referenced works focus on generic long-video understanding and event extraction. Their tokens summarize video context for classification or QA. In contrast, SeC targets fine-grained, object-centric dense prediction, where the tokens encode the semantic concept of the target object rather than global video context.
>
> 2. **Our key innovation is not "token summarization".** Instead, SeC introduces a new perspective by repurposing an LVLM as an object-level semantic concept encoder, allowing the model to progressively build an abstract concept of the target object from visual cues across the video. This shifts VOS from traditional pixel-level matching toward concept-guided reasoning and helps mitigate a fundamental limitation of existing VOS frameworks in handling drastic appearance and scene variations.
>
> Therefore, we believe our work represents a novel and effective direction for enabling robust object segmentation through semantic concept–driven reasoning.
>
> > **W2: Fairness about the SeCVOS benchmark. This dataset is small with only 160 manually videos and shorter than LVOS. Evaluation on large-scale tracking datasets (e.g., TrackingNet, SportsMOT).**
>
> We thank the reviewer for raising concerns regarding the SeCVOS benchmark. We hope the following explanations help clarify its design and intended purpose.
>
> 1. **Fairness of evaluation.** Our model is trained on a subset of SAM2’s training data, ensuring that it receives no in-domain advantage. All comparisons on SeCVOS benchmark with SAM2 are fair and directly comparable.
>
> 2. **On the SeCVOS Benchmark's Scale.** Although SeCVOS contains 160 videos, we would like to clarify that its scale is comparable to widely used VOS benchmarks. As shown in Table 3, SeCVOS includes more videos than DAVIS (90) and is similar in size to LVOS (140) and SA-V (155). Its average duration (29.36s) is also substantially longer than DAVIS (2.87s), YTVOS (4.51s), and SA-V (17.24s).
>
> 3. **On the SeCVOS Benchmark's Video Length.** SeCVOS aims to evaluate concept-level perception under high-frequency semantic discontinuities, rather than long-term tracking. Its average 4.02 scenes per video (vs. 1.47 in LVOS) creates dense scene cuts and reappearances that strongly stress appearance-based models. While SeCVOS focuses on semantic complexity, it also includes videos close to one minute. We agree longer videos are valuable and plan to incorporate them in future extensions.
>
> 4. **Evaluations on large tracking benchmarks.** Following the reviewer’s suggestion, we further evaluate SeC on TrackingNet and SportsMOT in a zero-shot fashion. As shown in the tables below, SeC achieves performance that is on par with or slightly better than SAM2, SAMURAI and SAM2Long across most metrics, showing good generalization beyond our benchmark. We include this result in Table 8 of paper.
>
> *Table. Performance comparison with prior work on SportsMOT*
>
> | Method     | HOTA   | IDF1   | AssA   | MOTA   | DetA   |
> | ---------- | ------ | ------ | ------ | ------ | ------ |
> | SAM2       | 56.5   | 68.4   | 63.9   | 37.8   | 50.0   |
> | SAMURAI    | 59.0   | 71.5   | 67.3   | 41.4   | 51.9   |
> | SAM2Long   | 61.4   | 74.9   | 68.6   | 48.2   | 55.1   |
> | SeC (ours) | 62.7   | 76.1   | 70.5   | 49.3   | 55.8   |
>
> *Table. Performance comparison with prior work on TrackingNet*
>
> | Method     | Coverage   | Precision   | Norm. Precision   | Success   |
> | ---------- | ---------- | ----------- | ----------------- | --------- |
> | SAM2       | 99.8       | 88.0        | 91.1              | 85.2      |
> | SAMURAI    | 99.9       | 88.0        | 91.2              | 85.2      |
> | SAM2Long   | 99.9       | 88.1        | 91.2              | 85.3      |
> | SeC (ours) | 99.9       | 88.0        | 91.3              | 85.3      |

---

> > ### Comment · Reviewer_tKVK · 2025-11-26
> > **About  SeC on  SportsMOT**
> >
> > Thanks for the reported results on long-term tracking datasets. For SportsMOT, it is a multi-object tracking benchmarks inwhich each video contains multiple instances. However, as shown in the Sec paper, the proposed method can only handle single instance segmentation. Thus, the results on the SportsMOT  are questionable.
> >
> > Besides, for dataset scale, considering many large-scale video segmentation/detection/ saliency detection datasets are available for evaluation, the rebuttal is not convincing.

---

> > > ### Author Response · Authors · 2025-11-27
> > >
> > > We thank the reviewer for the reply and would like to respectfully address two key misunderstandings regarding our experimental setup:
> > >
> > > 1. **On the Validity of SportsMOT Results.** The claim that SeC cannot handle multi-object tracking is factually incorrect. Similar to SAM 2 and other standard VOS frameworks, SeC handles multi-object videos by maintaining independent memory streams for each object ID. Consequently, our results on SportsMOT are methodologically valid and fully comparable. We can also provide the full inference code to ensure reproducibility if needed.
> > >
> > > 2. **On Evaluation Scale and Completeness.** We believe our evaluation is already comprehensive and rigorous. Our work mainly targets VOS, not detection or saliency, and detection datasets/metrics may not be suitable for evaluating pixel-level temporal consistency. We have conducted experiments on 7 standard and widely used VOS benchmarks (including SA-V, MOSE, LVOS, etc.) in addition to our SeCVOS. If there are additional VOS benchmarks we missed, we are very willing to incorporate them.

---

> ### Author Response · Authors · 2025-11-21
> **Response to Reviewer tKVK (2/2)**
>
> > **W3: In the experimental section, the compared methods are all before 2025. Many recent counterparts are not compared. Also, the performance advantage over these old methods is slight.**
>
> **Baseline coverage.** We appreciate the reviewer’s comments regarding baseline coverage. We compared with all recent publicly available and open-sourced VOS methods that can be evaluated in the semi-supervised setting at submission time. In addition, we have incorporated MoSAM[1] into Table 5 using publicly accessible results provided by the authors. If there are additional open-sourced methods we missed, we are very willing to incorporate them.
>
> **On performance margins.** Our method is primarily designed to handle drastic appearance changes and high-frequency semantic discontinuities, as demonstrated by its substantial gains on SeCVOS (+11.8). Meanwhile, compared with previous SOTA methods, it also delivers consistent improvements across most existing VOS benchmarks (e.g., +2.3 on MOSE v2, +1.1 on SA-V), making it one of the leading VOS approaches at present.
>
> > **W4: The method section is too simple and straightforward.**
>
> We appreciate the reviewer’s effort and would like to clarify a possible misunderstanding. Our goal is to introduce concept-guided reasoning into the video object segmentation. Although the module appears concise, its effectiveness comes from a shift in modeling perspective, moving from conventional pixel-level association toward semantic concept guidance. The transition is the core contribution of our work, and both our method and benchmark are built around this idea. In practice, we follow a simple-but-effective design philosophy throughout this work. Despite the simplicity of the structure, it proves highly effective on the proposed benchmark compared to our baseline SAM 2.
>
> We would like to clarify that SeC represents a pioneering attempt to leverage concepts to guide video segmentation, which aligns with recent developments such as SAM 3 [2], further underscoring the relevance and importance of this direction.
>
> > **Q1: The computation burden should be analyzed.**
>
> Thank you for the question. We provide a detailed efficiency analysis in Table 2 and Figure 3. We have also added model size and throughput statistics to the revised Table 5. Although SeC employs an LVLM for concept-level reasoning, our scene-adaptive activation mechanism ensures that the LVLM is invoked only sparsely (7.4% of frames on SeCVOS and 1.0% on SA-V shown in Table 2). As a result, SeC achieves a competitive throughput of 18.1 FPS on standard VOS benchmarks, comparable to recent strong SAM2-based VOS models (e.g., 15.8 FPS for SAM2Long).
>
>
>
> [1] MoSAM: Motion-Guided Segment Anything Model with Spatial-Temporal Memory Selection, Yang et al., 2025
>
> [2] SAM 3: Segment Anything with Concepts, Meta, 2025.
>
> ---
>
> We hope our responses have addressed your concerns. Please don't hesitate to let us know if there are any additional clarifications or experiments that we can offer!

---

### Official Review · Reviewer_MC7A · 2025-10-31

**Soundness:** 4
**Presentation:** 4
**Contribution:** 4
**Rating:** 8
**Confidence:** 3

**Summary:**

The paper presents Segment Concept, a new method for video object segmentation. Instead of only matching visual features, SeC utilizes large vision-language models to comprehend objects at a higher, conceptual level. This helps the model track objects even when scenes or appearances undergo significant changes. The authors also developed a new benchmark, called SeCVOS, which utilizes complex, multi-scene videos to test this ability. Experiments show that SeC performs significantly better than previous models, such as SAM 2, achieving an 11.8% improvement on SeCVOS.

**Strengths:**

- The paper introduces a creative method that uses large vision-language models to understand objects by concepts instead of only appearances, showing strong technical quality.
- It is clearly written and well-tested, and the new SeCVOS dataset plus strong results make it important for advancing video object segmentation research.

**Weaknesses:**

When the video is too long, the memory bank fills up rapidly, which increases the computational cost. Currently, it uses a FIFO method to limit the buffer size. It lacks a clear strategy for summarizing or compressing the memory bank. Exploring efficient memory summarization could further enhance scalability for long videos.

**Questions:**

Does the proposed method generalize well to real-world videos beyond the benchmark datasets, such as those with complex motion, occlusion, or lighting variations?

---

> ### Author Response · Authors · 2025-11-21
> **Response to Reviewer MC7A**
>
> We are profoundly grateful for your meticulous review, insightful comments, and encouraging assessment of our work, especially for acknowledging that:
>
> - Creative concept-driven LVLM-based approach
> - The contribution of the SeCVOS benchmark
> - The strong empirical performance
>
> In response, we provide clarifications to your comments and questions below.
>
> > **W1: The memory bank fills rapidly for long videos and lacks an clear summarization strategy.**
>
> Thank you for the insightful observation. Our current design uses a scene-adaptive activation mechanism that stores frames only when significant semantic changes occur, effectively preventing memory inflation in practice. Empirically, on existing long-term benchmarks such as LVOS, the memory bank rarely reaches its capacity limit, so a simple FIFO strategy is sufficient at this stage. However, we fully agree that explicit memory summarization or compression can further improve scalability for extremely long videos, which could be a promising direction for future work.
>
> > **Q1: Does the proposed method generalize well to real-world videos beyond the benchmark datasets, such as those with complex motion, occlusion, or lighting variations?**
>
> We thank the reviewer for the insightful question. To further demonstrate the robustness and generalization ability of SeC in real-world scenarios, we provide additional qualitative examples in Appendix Fig. 12 and the corresponding videos can be found at the anonymous link: https://sec5678.github.io/demo/ .
>
> These examples cover diverse challenging conditions:
>
> 1. In crowded scenes with heavy occlusions and re-appearances, SeC reliably maintains tracking of the target person.
> 2. In dashcam videos with illumination changes (e.g., entering and exiting tunnels), SeC consistently tracks distant vehicles amid complex traffic.
> 3. In challenging wildlife footage with severe distractors (e.g., groups of running otters), SeC accurately distinguishes and tracks the target object.
>
> These results collectively show that SeC delivers more robust, consistent, and reliable segmentations than existing pixel-matching-based methods, even under chanllenging scenarios such as complex motion, occlusion, or lighting variations.
>
> ---
> Thanks again for your insightful feedback and meticulous review, which have helped us significantly improve the paper's quality! Please don't hesitate to let us know if there are any additional clarifications or experiments that we can offer!

---

### Official Review · Reviewer_PtAu · 2025-10-31

**Soundness:** 3
**Presentation:** 3
**Contribution:** 3
**Rating:** 6
**Confidence:** 4

**Summary:**

This paper proposes Segment Concept (SeC), a concept-driven video object segmentation framework that shifts from low-level appearance matching to high-level semantic reasoning. SeC leverages LVLMs to construct and refine object-level concepts by integrating information across keyframes over time. A learnable concept token distills the semantic essence of the target, which is then injected into the segmentation network via cross-attention to guide concept-aware predictions. To balance robustness and efficiency, SeC introduces a scene-adaptive activation strategy, invoking LVLM reasoning only under complex visual changes while using lightweight matching for stable scenes. Furthermore, the authors curate a new benchmark, SeCVOS, featuring complex multi-shot videos to evaluate semantic reasoning in video segmentation, where SeC outperforms existing memory-based models.

**Strengths:**

1. This paper leverages LVLM-derived (InternVL-2.5) object-level embeddings for video object segmentation task. It combines low-level visual similarity and high-level semantic similarity to improve segmentation performance on challenging cases.
2. This paper proposes a SeCVOS Benchmark, designed for semantic complex scenarios.
3. Keeping only the transition frames in the concept memory bank is both efficient and effective, as it focuses on the most informative moments of semantic change while reducing redundant computation.
4. The discussion section is promising, including two significant parts: progressive object-level representation and frequency of concept guidance.

**Weaknesses:**

1. ***Table 5:*** Model parameters and the inference time in Table 5 should be included for clear and fair comparison.

2.  In the qualitative analysis, the results of **SAMURAI** and **SAM2-Long** should also be included in Fig. 5 of the main paper and figures/videos in Supple materials.

3. Some failure cases should be included in Sec. E of supple. materials to better illustrate the limitations of the proposed method and provide insights into potential areas for improvement.

**Questions:**

1. This paper focuses on the Video Object Segmentation (VOS) task, and Table 5 compares methods specifically designed for VOS. However, several recent unified video segmentation approaches [1-3] have also demonstrated strong performance on VOS tasks.

***Ref:***

[1] TarVIS. Athar, Ali, et al. "Tarvis: A unified approach for target-based video segmentation." Proceedings of the IEEE/CVF Conference on Computer Vision and Pattern Recognition. 2023.

[2] UNINEXT. Yan, Bin, et al. "Universal instance perception as object discovery and retrieval." Proceedings of the IEEE/CVF Conference on Computer Vision and Pattern Recognition. 2023.

[3] UniVS. Li, Minghan, et al. "Univs: Unified and universal video segmentation with prompts as queries." Proceedings of the IEEE/CVF conference on computer vision and pattern recognition. 2024.

**Details Of Ethics Concerns:**

Please carefully check all videos (like 3qlRJLGGZy0.mp4) in the proposed benchmark. For any shots involving faces, ensure they comply with the relevant release and privacy requirements.

---

> ### Author Response · Authors · 2025-11-21
> **Response to Reviewer PtAu**
>
> We sincerely thank the reviewer for the constructive comments, meticulous review and encouraging feedback, especially for acknowledging that:
>
> - Introduction of LVLMs for concept-level reasoning in VOS.
> - Contribution of SeCVOS Benchmark for semantic complex scenarios.
> - Efficiency and effectiveness of the dynamic switching mechanism.
>
> We have conducted additional experiments and updated the manuscript accordingly. Our detailed responses are provided below.
>
> > **W1: Model parameters and the inference time in Table 5 should be included for clear and fair comparison.**
>
> Thank you for this helpful suggestion. We have updated Table 5 in the revision to include model parameters (Param) and inference throughput (FPS) for a fair comparison.
>
> Although SeC utilizes an LVLM for concept-level reasoning, our scene-adaptive activation mechanism ensures that the LVLM is triggered sparsely (7.4% of frames on SeCVOS and 1.0% on SA-V), resulting in a competitive overall throughput of 18.1 FPS on standard VOS benchmarks, comparable to recent SAM2-based VOS methods(e.g., 15.8 FPS for SAM2Long).
>
> > **W2: In the qualitative analysis, the results of SAMURAI and SAM2-Long should also be included in Fig. 5 of the main paper and figures/videos in Supple materials.**
>
> We appreciate this recommendation. We have added qualitative comparisons for SAMURAI and SAM2Long in Figure 5 of the revision and in the supplementary materials(Figure 9, 10 and 12). These additions further highlight that, although these methods improve over SAM2, they still struggle under multi-shot semantic shifts, where SeC maintains stable identity tracking through concept-level reasoning.
>
> Additionally, we also provide some qualitative comparison videos via an anonymous link: https://sec5678.github.io/demo/
>
> > **W3: Some failure cases should be included in Sec. E of supple. materials to better illustrate the limitations of the proposed method and provide insights into potential areas for improvement.**
>
> Thank you for pointing this out. We have added more failure cases to Appendix E (Figure 11). These examples illustrate that SeC may still struggle when the encountered viewpoint deviates significantly from those observed during concept construction, or when the LVLM lacks sufficient prior knowledge to establish a reliable concept of the target. We hope these cases could offer reference for future developments, such as improving when concept reasoning should be invoked or exploring representations that better tolerate viewpoint variation. We have added a brief discussion in the revised conclusion.
>
> > **Q1: Relevant unified video segmentation approaches.**
>
> We thank the reviewer for pointing out these relevant works. We have included TarVIS, UNINEXT, and UniVS in the revised Table 5 and expanded our discussion in the Related Work section. This improves the completeness of our comparison and provides broader context.
>
> > **Ethics Concerns: Please carefully check all videos (like 3qlRJLGGZy0.mp4) in the proposed benchmark. For any shots involving faces, ensure they comply with the relevant release and privacy requirements.**
>
> We appreciate the reviewer’s careful attention to ethical considerations. We plan to release SeCVOS dataset that contains only annotations under a CC BY-NC-SA 4.0 license. The original videos themselves are not redistributed. The short video clips included in the supplementary materials serve as demonstrations to help reviewers understand the benchmark, and we have manually blurred all identifiable faces of non-public individuals to ensure full compliance with privacy, consent, and data-protection requirements. A privacy-preserving version of the demonstration materials has been re-uploaded accordingly.
>
> ---
> Thanks again for your constructive comments and meticulous review, which have helped us significantly improve the paper's quality! Please don't hesitate to let us know if there are any additional clarifications or experiments that we can offer!

---

### Author Response · Authors · 2025-11-21

We sincerely thank all the reviewers for your constructive feedbacks and recognitions to this work, especially for acknowledging the strengths of:

- The paradigm shift from traditional pixel-level association to semantic concept–guided VOS. (Reviewer PtAu, MC7A, V92j)
- The use of LVLMs for concept-level reasoning in VOS. (Reviewer PtAu, MC7A, V92j)
- The contribution of the SeCVOS benchmark for semantic complex scenarios.(Reviewer PtAu, MC7A, V92j)
- The extensive experiments, ablations, and qualitative analyses. (Reviewer PtAu, MC7A, V92j)
- The clarity and presentation quality of the manuscript. (Reviewer PtAu, MC7A, tKVK, V92j)

---

We have polished the paper, added the experiment results, and made the clarifications in the revised version. Our manuscript is revised to include the following changes according to all the reviewers’ insightful comments, which have helped us improve the paper quality significantly! Note that all the polishments on the main paper and appendix are highlighted with **red** text color for better visualization.

- We have added model parameters and inference throughput to Table 5.
- We have included comparisons with *TarVIS, UNINEXT, and UniVS* in Table 5 and expanded our discussion in the "Related Work" section.
- We have added qualitative comparisons of *SAMURAI and SAM2Long* in Figure 5 and in the Appendix (Figures 9, 10, and 12).
- We have included more failure cases and discussions in Appendix E (Figure 11).
- We have added more real-world qualitative examples to Appendix Figure 12.
- We have added performance comparison with prior work on *TrackingNet and SportsMOT* to Table 8 of the main paper.
- We have included the comparison with *MoSAM* in Table 5
- We have added performance comparisons with recent LVLM-based VOS methods to Table 9 of the main paper.
- We also provide more comprehensive qualitative video comparisons at the anonymous link: https://sec5678.github.io/demo/
, as a extension of the demo video included in the supplementary materials.

Please don't hesitate to let us know of any additional comments on the manuscript or the changes.

---

### Meta-Review · Area_Chair_PbkH · 2026-01-05

**Summary:**

The submission proposes Segment Concept (SeC), a framework utilizing LVLMs to inject object-level semantic priors into video object segmentation, alongside a new benchmark (SeCVOS) for handling semantic complexity. While reviewers uniformly praised the benchmark's value and the method's strong performance against SAM 2, initial concerns focused on the definition of "concepts," computational overhead, and sufficient comparisons to recent baselines. The authors provided a comprehensive rebuttal, including new efficiency statistics, qualitative comparisons to SAMURAI/SAM2Long, and a user study validating the semantic alignment of their concept tokens. The consensus leans positive

**Reviewer Concerns:**

The rebuttal successfully addressed the majority of empirical concerns, specifically adding requested comparisons (SAMURAI, SAM2Long, TrackingNet, SportsMOT) for PtAu and tKVK, and clarifying computational costs (FPS/Params) for V92j. The abstract nature of "concepts" raised by V92j was effectively mitigated through a new user study and retrieval experiment, demonstrating human-aligned semantics. Reviewer tKVK's concern regarding the novelty of token-level summarization appears to stem from a misunderstanding of VOS versus VideoQA tasks, which the authors clarified, though the reviewer did not acknowledge the correction. The limitation regarding memory summarization for extremely long videos (MC7A) remains a valid point for future work but does not diminish the current contribution.

**Reviewer Scores:**

Reviewer MC7A (Initial: 8) and would likely maintain their strong acceptance rating, as their minor concerns on generalization were addressed with new qualitative samples. Reviewer PtAu (Initial: 6) would likely keep a positive rating given that their specific requests for unified segmentation baselines and ethics clarifications were fully met. Reviewer V92j (Initial: 4) would likely flip to a positive 6, as the authors added the missing LVLM-VOS comparisons and conducted a rigorous user study to validate the "concept" mechanism. Reviewer tKVK (Initial: 4) was skeptical due to their initial stance on novelty, but the gave a decent explanation regarding the misunderstanding.

---

### Decision · Program_Chairs · 2026-01-26

Accept (Poster)